# Rethinking and Improving Autoformalization: Towards a Faithful Metric and a Dependency Retrieval-based Approach

**Qi Liu, Xinhao Zheng, Xudong Lu, Qinxiang Cao[∗], Junchi Yan[∗†]**
Sch. of Computer Science & Sch. of Artificial Intelligence, Shanghai Jiao Tong University
{purewhite,void_zxh,luxudong2001,caoqinxiang,yanjunchi}@sjtu.edu.cn
https://github.com/Purewhite2019/rethinking_autoformalization

## Abstract

As a central component in formal verification, statement autoformalization has been widely studied including the recent efforts from machine learning community, but still remains a widely-recognized difficult and open problem. In this paper, we delve into two critical yet under-explored gaps: 1) absence of faithful and universal automated evaluation for autoformalization results; 2) agnosia of contextual information, inducing severe hallucination of formal definitions and theorems. To address the first issue, we propose **BEq** (*Bidirectional Extended Definitional Equivalence*), an automated neuro-symbolic method to determine the equivalence between two formal statements, which is formal-grounded and well-aligned with human intuition. For the second, we propose **RAutoformalizer** (*Retrieval-augmented Autoformalizer*), augmenting statement autoformalization by *Dependency Retrieval*, retrieving potentially dependent objects from formal libraries. We parse the dependencies of libraries and propose to *structurally informalise* formal objects by the topological order of dependencies. To evaluate OOD generalization and research-level capabilities, we build a novel benchmark, *Con-NF*, consisting of 961 informal-formal statement pairs from frontier mathematical researches. Experiments validate the effectiveness of our approaches: BEq is evaluated on 200 diverse formal statement pairs with expert-annotated equivalence label, exhibiting significantly improved accuracy ($82.50\% \mapsto 90.50\%$) and precision ($70.59\% \mapsto 100.0\%$). For dependency retrieval, a strong baseline is devised. Our RAutoformalizer substantially outperforms SOTA baselines in both in-distribution ProofNet benchmark ($12.83\% \mapsto 18.18\%$, BEq@8) and OOD Con-NF scenario ($4.58\% \mapsto 16.86\%$, BEq@8).

> *Philosophy is written in this grand book, the universe.*
> *It is written in the language of mathematics.*

> Galileo Galilei, *The Assayer*

## 1 Introduction

Theorem provers, such as Lean (Moura & Ullrich, 2021), Coq (Bertot & Castéran, 2013) and Isabelle (Nipkow et al., 2002), can check the validity and correctness of mathematical statements and proofs by strict algorithms, whose own soundness and completeness are proven in theory. However, instead of directly working on natural language mathematics, these tools define their own formal languages, which hinders the democratization of formal mathematics.

Statement autoformalization aims at translating mathematical statements from natural language to formal verifiable statement. Readers unfamiliar with formal theorem proving are advised to read Yang et al. (2024). Due to its rigorously logical nature, this task is widely-recognized to be challenging, requiring profound understanding of both informal semantics and formal syntax (Li et al., 2024a). Beyond a fundamental component in formal mathematics and software verification, strong autoformalization methods have far broader impacts and could result in the creation of a general

---

[∗]Equal correspondence. [†]Also affiliated with Shanghai Artificial Intelligence Laboratory. This work was in part supported by NSFC (92370201, 62222607) and Shanghai Municipal Science and Technology Major Project under Grant 2021SHZDZX0102.

purpose reasoning module (Szegedy, 2020). Outside-the-box applications of autoformalization include synthesizing training dataset for formal theorem provers (Wu et al., 2022; Xin et al., 2024), especially AlphaProof (Castelvecchi, 2024), enhancing informal math reasoning by rejection sampling (Zhou et al., 2024), and automating code verification (Lin et al., 2024).

Current mainstream methods work in the following process. A large language model (LLM) is either prompted (Wu et al., 2022) or fine-tuned (Azerbayev et al., 2023; Jiang et al., 2023a) to directly generate a formal statement given its informal counterpart. The predicted statements are then evaluated by laborious human annotation (Azerbayev et al., 2023) or unreliable proxy automated metrics including machine translation metrics such as BLEU (Wu et al., 2022) and perplexity (Wang et al., 2018), symbolic type check pass rate (Lu et al., 2024) or LLM grader (Ying et al., 2024a).

Rethinking this paradigm, we find out two key limitations. Firstly, an **effective, human-aligned and universal automated evaluation metric is absent**. Machine translation metrics are fragile to equivalent transformations in human perspective, for example $\beta$-*reduction* (function application). Type check is too weak to filter out syntactically correct but semantically absurd autoformalization. It is a necessary but not sufficient condition for the ideal equivalence. LLM graders are non-determinant and highly dependent on prompts, and are easily misled by imperceptible but fundamental differences or huge but nonessential transformations. Murphy et al. (2024) are pioneers to utilize SMT solver for faithful automated evaluation, but is restricted to Euclidean geometry only. Secondly, the current paradigm directly generates formal statements, **ignoring the context of previously formalized statements and definitions**. This might result in severe hallucination of identifiers and syntax, especially in out-of-distribution (OOD) cases. A similar issue is reported in Wu et al. (2022), where definition misalignment between informal mathematics and formal libraries is the major cause of failure cases. Our experiments on both in-domain and OOD scenarios, shown in Table 3, show the severity of this problem and exhibit a promising path to address it.

For the first issue, we propose *BEq (Bidirectional Extended Definitional Equivalence)*, a neural-symbolic equivalence relation between formal statements. This metric aligns well with collective human opinions. In formal systems built upon dependent type theory (Univalent Foundations Program, 2013), such as Lean 4 (Moura & Ullrich, 2021), definitional equality is a symbolic equivalence relation under a variety of intuitive transformations, such as bound variable renaming, function application, and definition unfolding. However, it heavily relies on the definitions of objects and conversion rules, hence it is too strict and inflexible from human perspective. For example, `n + 0` and `n` are definitional equal for a natural number $n$, but `n` and `0 + n` are not. Worse still, definitional equality struggle with handling metavariable differences. We extend definitional equivalence by 1) equipping it with a restricted set of symbolic transformation primitives and a neural transformation function aiming to convert one formal statement to be definitionally equivalent to the other, and 2) loosing the equivalence criteria to bidirectionally "convertible" under the transformation function. To evaluate its performance, we build a benchmark consisting of 200 formal statement pairs with expert-annotated equivalence labels. BEq significantly outperforms previous SOTA methods, improving the precision from $70.59\%$ to $100\%$ and the accuracy from $82.50\%$ to $90.50\%$ .

For the second, we propose a new task, *Dependency Retrieval*, and a new method, *RAutoformalizer (Retrieval-augmented Autoformalizer)*. Dependency retrieval seeks to select potentially dependent formal objects given an informal statement. RAutoformalizer uses the retrievals to enhance autoformalization. To enable this new paradigm, we propose to parse the dependencies in formal libraries and construct training data by *topological informalization*, informalizing formal objects by topological order. An immense dataset of 243,797 formal objects (including 139,933 theorems) is synthesized upon Mathlib 4. We also build the *Con-NF* benchmark[1] to evaluate out-of-distribution (ODD) generalization and research-level capabilities of current methods. A baseline is built for dependency retrieval, with $35.52\%$ Recall@5 on ProofNet and $24.32\%$ Recall@5 on Con-NF. RAutoformalizer exhibits substantial improvement over previous methods, improving BEq@8 from $12.83\%$ to $18.18\%$ on ProofNet and from $4.58\%$ to $16.86\%$ on Con-NF.

To sum up, we identify two key limitations in statement autoformalization: 1) absence of faithful and universal automated evaluation; 2) agnosia of contextual information. The contributions are:

1) We give a neural-symbolic equivalence metric, **BEq** (*Bidirectional Extended Definitional Equivalence*), extending *Definition Equality* in dependent type theory more aligned with human intuition.

---

[1]Based on Lean 4 Con(NF) library (A formal consistency proof of Quine's set theory *New Foundations*)

2) We propose a new *dependency retrieval* task and introduce a novel paradigm, **RAutoformalizer** (Retrieval-Augmented Autoformalizer). We further propose *topological informalization* to synthesize high-quality training data for these initiatives. To evaluate research-level autoformalization and out-of-distribution (OOD) performance, we create a new benchmark, *Con-NF*, which consists of 961 informal-formal statement pairs from New Foundations (Holmes & Wilshaw, 2024).

3) We validate BEq by expert evaluation on 200 formal statement pairs and set a baseline for dependency retrieval. Extensive experiments of RAutoformalizer show its superior performance on statement autoformalization. Ablation studies further validate the effectiveness of our technical modifications, and also exhibit the great potential of the retrieval-augment paradigm.

## 2 RELATED WORKS

**Autoformalization.** It aims to automatically translate the natural language (informal) mathematics into formal verified code. Current autoformalization methods can be roughly divided into three levels. Statement autoformalization focuses on autoformalizing statements (Wang et al., 2020; Wu et al., 2022; Azerbayev et al., 2023; Jiang et al., 2023a; Gulati et al., 2024; Poiroux et al., 2024); proof autoformalization focuses on translating informal proofs (and sometimes including corresponding statements) into formal code (Cunningham et al., 2023; Jiang et al., 2023b; Zhao et al., 2023; Murphy et al., 2024; Lu et al., 2024); theory autoformalization, translating a whole theory including definitions, axioms, theorems, and proofs, remains under-explored. Patel et al. (2024) proposes a three-stage plan to break the difficulty into easier subtasks.

**Methods of Autoformalization.** Autoformalization is notoriously challenging for prevalent data-driven approaches (Li et al., 2024b). Existing informal-formal parallel corpora are fairly scarce, which impedes machine learning training. To alleviate this, researchers synthesize informal-formal pairs by rule-based informalization (Wang et al., 2018; Cunningham et al., 2023), LLM-based back-translation (Azerbayev et al., 2023; Jiang et al., 2023a), training with multilingual corpus (Jiang et al., 2023a), or utilizing in-context learning (ICL) capabilities (Wu et al., 2022). Ying et al. (2024a) proposes an expert iteration pipeline by iteratively synthesizing and filtering training data.

A major difference from machine translation is the existence of verifiers. Another line of work focuses on utilizing verifier feedbacks. Poiroux et al. (2024) uses rejection sampling to enhance autoformalization by typecheck results; Lu et al. (2024) introduces a neural step-level verifier and perform expert iteration; Jiang et al. (2023b); Murphy et al. (2024) combines LLM and formal verifier for proof autoformalization, and Zhao et al. (2023) enhances it with subgoal-based demonstration.

**Evaluation of Autoformalization.** There are many benchmarks for statement autoformalization, covering undergraduate-level math problems (Azerbayev et al., 2023), more complex areas from Mathlib 4 (Gulati et al., 2024), and Euclidean geometry (Murphy et al., 2024).

Due to the high flexibility of natural language and the rigor of formal language, faithfully evaluating autoformalization is widely-recognized to be challenging and under-explored (Szegedy, 2020; Azerbayev et al., 2023; Jiang et al., 2023a; Murphy et al., 2024). Wu et al. (2022); Jiang et al. (2023a); Ying et al. (2024a) evaluate autoformalization results by human experts. Wang et al. (2018) reports identical matching accuracy. Proxy metrics, including perplexity (Wang et al., 2018), BLEU[2] (Wang et al., 2018; Poiroux et al., 2024; Azerbayev et al., 2023; Wu et al., 2022) and compiler typecheck pass rate (Lu et al., 2024; Azerbayev et al., 2023; Jiang et al., 2023a) are utilized to automate evaluation. Ying et al. (2024a); Gulati et al. (2024) prompts LLMs to determine the equivalence between predicted formal statement and ground-truth. Murphy et al. (2024) propose to use SMT solver to evaluate the equivalence between formal statements in Euclidean geometry.

For proof autoformalization, current evaluation focuses on theorem proving, only verifying formal proofs' correctness while potentially overlooking semantic inconsistencies between informal and formal proofs. The evaluation of theory autoformalization is also insufficiently researched.

**Retrieval-augmented Generation.** It has been extensively studied in NLP. For code generation, code documentations (Zhou et al., 2023), APIs (Zan et al., 2022), repository files (Zhang et al., 2023) and dynamic knowledge soup (Su et al., 2024) are retrieved to augment generation. In formal verification, Azerbayev et al. (2023) proposes to augment statement autoformalization by retrieving relevant prompts. ReProver (Yang et al., 2024) enhances theorem proving with premise selection.

---

[2] BLEU (Papineni et al., 2002) is a metric for evaluating machine translation based on n-gram matching.

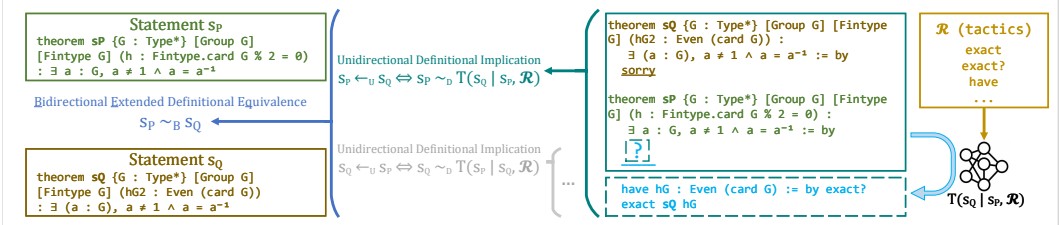

Figure 1: Illustration of *BEq* (*Bidirectional Extended Definitional Equivalence*) and *Unidirectional Definitional Implication*. $s_P \sim_B s_Q$ if and only if both $s_P \leftarrow_U s_Q$ and $s_Q \leftarrow_U s_P$ hold. To determine the first, we assume $s_Q$ holds. Then the transformation function (implemented with a LLM) $T$ is called to generate transformation (proof of $s_P$ using $s_Q$) conditioned on $s_Q$ and transformation primitive (tactic) set $\mathcal{R}$. If the transformation holds, we conclude that $s_P \leftarrow_U s_Q$. Otherwise, we believe $s_P \nleftarrow_U s_Q$. Vice versa for the second direction.

## 3 BIDIRECTIONAL EXTENDED DEFINITIONAL EQUIVALENCE

### 3.1 BACKGROUND

A fundamental problem for all generative tasks is to faithfully, effectively, and interpretably evaluate the results. In statement autoformalization, we follow prevalent benchmarks such as ProofNet (Azerbayev et al., 2023) and LeanEuclid (Murphy et al., 2024) to evaluate by comparing model predictions with ground-truths: let $\mathbb{S}$ denote the set of all formal statements, given a predicted formal statement $s_{\text{pred}} \in \mathbb{S}$ and the corresponding ground-truth $s_{\text{gt}} \in \mathbb{S}$, an equivalence relation $\sim: \mathbb{S} \times \mathbb{S}$ used to determine the correctness of autoformalization should be:

- $(\cdot \sim \cdot)$ equivalence relation: a binary relation with reflexivity, symmetry and transitivity.
- $(\cdot \sim \cdot)$ is well aligned with human intuition.
- $(\cdot \sim \cdot)$ is universally applicable in all domains.

Intuitively, equivalence from a human perspective is generally one that can be quickly determined and reasoned. Hence, the key lies in defining an equivalence relation that can be demonstrated through brief proofs. We choose to build an equivalence relation that aligns with humans by 1) extending *definitional equality*, and 2) restricting the degree of proof automation.

**Definitional Equality**. In Lean 4 (Moura & Ullrich, 2021), two expressions are *definitionally equal* if they are equivalent w.r.t. a series of conversion rules, such as $\alpha$-*conversion* (renaming bound variable), $\eta$-*expansion* (modifying unused arguments in functions), *proof irrelevance* (proofs of the same `Prop`), $\beta$-*reduction* (function application), $\zeta$-*reduction* (eliminating let-in definitions), $\delta$-*reduction* (unfolding variable and constant definitions), $\iota$-*reduction* (application of recursive functions defined on inductive types to an explicit constructor) (Bailey et al., 2024). This equality is a binary relation with reflexivity, symmetry, and transitivity, and it is applicable in all math areas formalized in Lean 4. And it has many intriguing characteristics that fit more closely with human instinct. For example, `fun (b:Nat) => b` is equivalent to `fun (u:Nat) => u` because definitional equality allows $\alpha$-conversion, in which bound variable `b` is renamed to `u`.

However, several critical weaknesses hinder definitional equality from becoming a good and intuitive metric for autoformalization. Firstly, some expressions that are naturally "equivalent" from a human perspective are not definitionally equal. For example, for a natural number `n:Nat`, `n + 0` and `n` are definitionally equal, but `0 + n` and `n` are not definitionally equal. Definitional equality heavily relies on the definitions of objects and conversion rules, while many intuitive equivalences, are neglected. Worse still, typecheck often get stuck in typeclass instance problems due to metavariables, which hinders evaluating definitional equality between statements.

### 3.2 EXTENDING DEFINITIONAL EQUALITY

**Formulation.** Suppose there are two formal statements, $s_P$ and $s_Q$. Without loss of generality, $s_P$ and $s_Q$ are assumed syntactically valid, since it is nonsense to talk about equivalence between invalid formal statements. Definitional equality is denoted as $\sim_D$.

The main reason behind the aforementioned limitations of definitional equality is its strictness on reductions and conversions. We hence loose the limitation and extend definitional equality to align

with human intuition. Let $\mathbb{R}$ be the set of all transformation primitives, $\mathcal{U}(s, \mathcal{R}) : \mathbb{S} \times 2^{\mathbb{R}} \mapsto 2^{\mathbb{S}}$ to be the set of all valid formal statements that can be constructed by applying transformations in $\mathcal{R} \subset \mathbb{R}$ on $s$, and $T : (\mathbb{S} \times (\mathbb{S} \times 2^{\mathbb{R}})) \mapsto \mathbb{S}$ to be a *restricted transformation function* such that

$$T(s_P | s_Q, \mathcal{R}) = \begin{cases} s'_P, & s'_P \in \mathcal{U}(s_P, \mathcal{R}) \wedge s_Q \sim_D s'_P \\ \bot, & \forall s'_P \in \mathcal{U}(s_P, \mathcal{R}), s_Q \not\sim_D s'_P \end{cases} \tag{1}$$

Intuitively, given transformation primitives $\mathcal{R} \subset \mathbb{R}$, $T$ transforms $s_P$ definitionally equal to $s_Q$ if possible and returns the transformed statement. Otherwise, it returns a dummy statement $\bot$, which is not definitionally equal to any other valid statement (e.g., an invalid statement).

In Lean 4, a formal statement can be converted to a proof goal by entering tactic mode. A proof goal $(\{s_{P,i}\}_{i=1}^n, s_Q)$ consists of some assumptions $\{s_{P,i}\}_{i=1}^n$ and a conclusion $s_Q$, where all $s_{P,i}$ and $s_Q$ are statements, and $n$ can be 0. Then tactics, which are metaprograms, reduce one goal to another, which is often easier to solve by assumptions. For example, transforming $(\{S\}, R \rightarrow S)$ to $(\{R, S\}, S)$ by tactic `intro` and trivially prove it by `exact`. A formal statement $s_P$ can be transformed to a proof goal by simply setting assumptions to be empty set and conclusion to be $s_P$, resulting in the proof goal $(\emptyset, s_P)$. And a proof goal $(\{s_{P,i}\}_{i=1}^n, s_Q)$ can be transformed back to a formal statement $s_{P,1} \wedge s_{P,2} \wedge \cdots \wedge s_{P,n} \rightarrow s_Q$. These transformations occur at the syntax level, leaving semantics unchanged. Therefore, we can determine semantic equivalence in the space of proof goals and concretize $\mathbb{R}$ to be the set of all tactics in Lean. The restricted transformation function $T$ can be approximated by sampling tactic sequences from a large language model and symbolically executing on Lean kernel multiple times, until a valid $s'_P$ is found, or the time limit exceeds. With a slight abuse of notation, we denote both the formal statement $s_P$ and its corresponding proof goal as $s_P$.

Then, *Unidirectional Definitional Implication* $(\cdot \leftarrow_U \cdot)$ is defined as

$$s_P \leftarrow_U s_Q \iff s_P \sim_D T(s_Q | s_P, \mathcal{R}) \tag{2}$$

Intuitively, this implication from $s_Q$ to $s_P$ indicates whether the proof goal of the statement $s_P$ can be definitionally equal to a restrictively transformed $s_Q$ by $T$. Correspondingly, **BEq** (*Bidirectional Extended Definitional Equivalence*) $(\cdot \sim_B \cdot)$ is defined as

$$s_P \sim_B s_Q \iff s_P \leftarrow_U s_Q \wedge s_Q \leftarrow_U s_P \tag{3}$$

which is

- a superset of definitional equality: Let $\mathcal{R} = \emptyset$, then, $T$ becomes identity mapping $\Delta(\cdot)$ and

$$s_P \sim_B s_Q \iff s_P \sim_D \Delta(s_Q) \wedge s_Q \sim_D \Delta(s_P)$$
$$\iff s_P \sim_D s_Q$$

- an equivalence relation, which is a binary relation with
  1. Reflexivity: $s_P \sim_B s_P$ holds because $s_P \sim_D s_P$.
  2. Symmetry: $s_P \sim_B s_Q \iff s_Q \sim_B s_P$ holds by unfolding the definition of BEq.
  3. Transitivity: If $s_P \sim_B s_Q$ and $s_Q \sim_B s_R$ holds, we have $s_P \sim_D T(s_Q | s_P, \mathcal{R})$ and $s_Q \sim_D T(s_R | s_Q, \mathcal{R})$. Suppose $T(s_Q | s_P, \mathcal{R})$ applies tactic sequence $[t_{QP}^{(i)}]_{i=1}^m$ to transform proof goal $s_Q$ to be definitionally equal to $s_P$, and $T(s_R | s_Q, \mathcal{R})$ applies $[t_{RQ}^{(j)}]_{j=1}^n$. Therefore, by applying $\text{Concat}([t_{RQ}^{(j)}]_{j=1}^n, [t_{QP}^{(i)}]_{i=1}^m)$ on $s_R$, we can transform proof goal $s_R$ to be definitionally equal to $s_P$. Therefore, $s_P \sim_D T(s_R | s_P, \mathcal{R})$.

**Implementation.** An overview of BEq is depicted in Figure 1. To implement the transformation function $T$, we perform 5-shot prompting InternLM-Math-Plus-20B (Ying et al., 2024b) served on vLLM (Kwon et al., 2023). If not mentioned otherwise, model prediction is sampled by beam search where temperature $T = 0.0$, attempt number $n = 8$, and beam size $b = 8$. The choice of transformation primitives is sophisticated and critical for aligning with humans. We set $\mathcal{R} = \{\texttt{apply},$ `cases'`, `constructor`, `exact`, `exact?`, `ext`, `have`, `intro`,`intros`, `rw`, `use`} to extend vanilla definitional equality (for higher recall) while preventing $\mathcal{U}(\cdot, \mathcal{R})$ and the equivalence class being too large (for higher precision). More experiments on the choices of attempt numbers, transformation primitives, and sampling strategies can be found in Appendix A.1.

Table 1: Comparison of automated evaluation metrics for statement autoformalization. **R**, **S**, **T** denote reflexivity, symmetry, and transitivity, respectively. **Universal** indicates whether a metric is applicable in all domains; 0/0 denotes division by zero; **I** and **D** denote InternLM2-Math-Plus-20B and Deepseek-V2.5, respectively; $\sim$ represents the metric is unsuitable for the method. *We report the best results among all thresholds; †Reflexivity and symmetry depends on the implementation.

| Metric | Binary Relation | | | Alignment with Human | | | Universal |
|---|---|---|---|---|---|---|---|
| | R | S | T | Precision↑ | Recall↑ | Accuracy↑ | |
| Identity Match | ✓ | ✓ | ✓ | 0/0 | 0.00% | 65.00% | ✓ |
| Typecheck | | $\sim$ | | 35.00% | 100.00% | 35.00% | ✓ |
| BLEU Threshold | ✓† | ×† | × | 62.96%* | 24.29%* | 68.50%* | ✓ |
| Majority Voting (I) | × | × | × | 40.00% | 94.29% | 48.50% | ✓ |
| Majority Voting (D) | × | × | × | 70.59% | 85.71% | 82.50% | ✓ |
| Definitional Equality | ✓ | ✓ | ✓ | 100.00% | 11.43% | 69.00% | ✓ |
| E3 (Murphy et al., 2024) | ✓ | ✓ | ✓ | | $\sim$ | | × |
| **BEq** | ✓ | ✓ | ✓ | 100.00% | 72.86% | 90.50% | ✓ |

Given two formal statements $s_P$ and $s_Q$, we first check $s_P \leftarrow_U s_Q$. $s_Q$ is assumed to be true by closing its proof with `sorry`. Then, symbolic heuristic `exact?` is called to generate a proof for $s_P$. If it fails, $n$ candidates are sampled from the LLM[3], given tactic restriction $\mathcal{R}$ and $s_Q$. If there exists at least one successful proof that uses $s_Q$, $s_P \leftarrow_U s_Q$ holds. Otherwise, $s_P \leftarrow_U s_Q$ does not hold. Then $s_Q \leftarrow_U s_P$ is similarly checked. If and only if both directions hold, $s_P \sim_B s_Q$ holds.

## 3.3 EVALUATION OF BEQ

**Human Equivalence Benchmark.** To fairly and reliably evaluate BEq and baseline metrics, we uniformly sampled 200 formal statements from the typechecked predictions generated by RAutoformalizer and OpenAI o1-preview (100 predictions from each). Then the statements are paired with the ground-truths in ProofNet (Azerbayev et al., 2023)[4]. Experts in math and formal verification are invited to discuss and label the equivalence in their opinion for the 200 statement pairs. The discipline distribution of these samples is visualized in Appendix A.4.

**Experiment Setting.** In our evaluation, identical matching is optimized to neglect spaces in formal statements. BLEU computation is identical to Azerbayev et al. (2023). To determine pairwise equivalence, we binarize BLEU by a threshold. The best results over all possible thresholds are reported. Precision, recall, and accuracy curves of different thresholds can be found in Appendix A.5. For LLM grader, we use the prompts[5] in Ying et al. (2024a) but a stronger setting: InternLM2-Math-Plus-20B (Ying et al., 2024b) and DeepSeek-V2.5 (DeepSeek-AI, 2024) with 16-shot majority voting and temperature $T = 0.7$. E3 (Murphy et al., 2024) is not evaluated, as it is only available in Euclidean Geometry. BEq also samples 16 tactic sequences candidates for each sample.

**Experiment Results.**[5] As summarized in Table 1, BEq reaches 100.0% precision and 90.50% accuracy, showing landslide advantages over baselines. However, BEq falls short on recall ($-12.85\%$ compared with "Majority Voting (D)") because of 1) rigor of formal verification systems; and 2) failure of approximated transformation function (the LLM), as analyzed in Appendix A.4. For baselines, Azerbayev et al. (2023) concludes that BLEU has a low correlation with ground-truth accuracy, with which our experiment result agrees. The distribution of BLEU scores of equivalent and inequivalent pairs is visualized in Appendix A.5. LLM Majority Voting sets a strong baseline, reaching 82.50% accuracy, but at the expense of precision. As a subset of BEq, definitional equality performs well in precision but has too many false negatives. With BEq, we can better evaluate statement autoformalization. In the following, we will address the second issue, agnosia of context.

## 4 RETRIEVAL-AUGMENTED AUTOFORMALIZATION

The current autoformalization paradigm suffers from the agnosia of context. Autoformalizers, without a priori knowledge of previously formalized definitions and theorems, frequently hallucinate

---

[3]Detailed prompt template can be found in Appendix A.7.

[4]All relevant open-source libraries are summarized in Appendix A.9.

[5]More comprehensive results can be found in Appendix A.2.1.

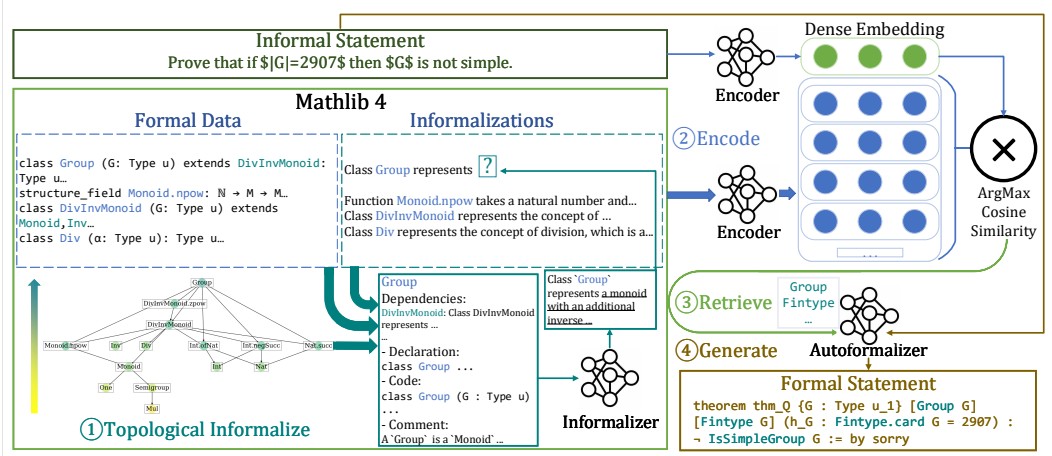

Figure 2: *RAutoformalizer*. **Train**: ①Dependencies in a library (e.g. Mathlib 4) are parsed. Formal objects are informalized by topological order, each given its own and dependencies' information. The resulting parallel data is used to train the retriever (encoder) and autoformalizer. **Inference**: ②Each informal statement is encoded to an embedding, whose cosine similarities are computed with pre-computed library embeddings. ③Objects corresponding to top-$k$ similarities are retrieved. ④Conditioned on the informal statement and retrieved dependencies, it predicts formal statements.

formal objects that are nonexistent in the library. This drawback is also observed as *definition misalignment* by Wu et al. (2022); Azerbayev et al. (2023); Jiang et al. (2023a). Although these hallucinated identifiers and function applications are semantically correct from the human perspective, formal verification fails because of the soundness of symbolic verifiers. Our preliminary experiments support this observation, with hallucination worsening in OOD scenarios like frontier research.

## 4.1 RAUTOFORMALIZER

We propose **RAutoformalizer** (*Retrieval-Augmented Autoformalizer*), which addresses the issue by incorporating *dependency retrieval*, selecting relevant formal objects for a given informal statement.

**Dependency Retrieval.** Suppose we are autoformalizing an informal statement $l_P$ with a ground-truth formal statement $s_P$ with dependencies $D_P$ from a formal library $\mathbb{D}$ (e.g., Mathlib 4). Dependency retrieval aims to retrieve a subset of formal objects $D$, maximizing the number of dependent formal objects of $s_P$ in $D$ while minimizing the inclusion of irrelevant ones, i.e.,

$$\arg \max_{D \in 2^{\mathbb{D}}} |D \cap D_P| - |D \cap D_P^{\complement}| \qquad (4)$$

Our retriever, $\psi_{\theta} : \mathbb{S} \mapsto \mathbb{S}^h$, which embeds a string onto the surface of a $h$-dimensional unit sphere, uses Dense Retrieval (Karpukhin et al., 2020) for its popularity, simplicity, and efficiency. Before inference, the embeddings of the whole library are precomputed as $\{\psi_{\theta}(s_d) \mid s_d \in \mathbb{D}\}$. Then, when an informal statement $l_P$ is provided, we only need a single forward pass to embed it as $\psi_{\theta}(l_P)$ and retrieve formal objects with top-$k$ maximal cosine similarities, see Figure 2 (Upper Right).

$$D = \arg \max_{D \in 2^{\mathbb{D}}, |D| \le k} \sum_{s_d \in D} \langle \psi_{\theta}(l_P), \psi_{\theta}(s_d) \rangle \qquad (5)$$

**Dataset.** We build the dependency graphs for Mathlib 4, illustrated in Figure 2 (Bottom Left), by parsing the declarations of all formal objects and linking identifiers with accessible formal objects in the corresponding context. In total, 243,797 formal objects (including 139,933 theorems) are collected along with their full names, positions, types, declarations, code, comments, and dependencies.

We propose to topologically informalize Mathlib 4 to synthesize a training dataset. Concretely, all formal objects are topologically sorted and split into 24 topological generations based on their dependency graph. Informalization is performed from the bottom (e.g., basic definitions) to the top (more sophisticated concepts), as Figure 2 (Bottom Middle) shows. We use 10-shot prompted

InternLM2-Math-Plus-20B (Ying et al., 2024b) as the informalizer. For a formal object, the informalizer is provided with the object's declaration, code[6], comment, and its dependencies' informalizations. The high quality of informalizations is shown in subsequent experiments.

**RAutoformalizer.** Building upon dependency retriever $\psi_{\theta}$, an LLM $p_{\phi}$ can predict formal statements given informal statements and retrieval results, as in Figure 2 (Bottom Right):

$$\hat{s}_P \sim p_{\theta}(\cdot | l_P, D) \tag{6}$$

The retriever $\psi_{\theta}$ is fine-tuned from BGE-M3 (Chen et al., 2023) using informalized theorems and dependencies in Mathlib 4 and hyperparameters in Appendix A.8. We retrieve the top-100 candidates using pretrained BGE-M3, remove true dependencies, and take the remainings as hard negatives (Xiao et al., 2023). By default, formal declarations of objects are used to generate embeddings.

For each theorem object, top-5 retrievals of $\psi_{\theta}$ are collected to fine-tune the autoformalizer $p_{\phi}$ from InternLM2-Math-Base-7B (Ying et al., 2024b) using the training recipe in Appendix A.8.

During inference, given an informal statement $l_P$ and a formal library $\mathbb{D}$, the retriever $\psi_{\theta}$ selects top-5 candidates from the library, then the autoformalizer $p_{\phi}$ generates formal statements based on the informal statement and retrievals.

**Con-NF: OOD Benchmark.** Existing benchmarks (Azerbayev et al., 2023; Zheng et al., 2022; Tsoukalas et al., 2024; Liu et al., 2023; Murphy et al., 2024) rely on Mathlib 4 and concentrate on high-school or undergraduate level mathematics. To evaluate the out-of-distribution generalization capabilities and research-level mathematics, we build a novel benchmark, *Con-NF*, based on Lean 4 Con(NF) (Holmes & Wilshaw, 2024) library. Con(NF) is a recently published digitization of Randall Holmes' proof (Holmes, 2015) that Quine's *New Foundations* (Quine, 1951) is consistent. We parse dependencies in this library, topologically informalize all 85,762 formal objects, deduplicate theorems from Mathlib 4, and eliminate unused formal objects of the remaining theorems. The cleaned benchmark consists of 961 theorems based on a different theoretical basis to merely Mathlib 4, along with a total of 1,348 formal objects and their informalizations.

## 4.2 Evaluation of Retrieval and Autoformalization

**Dependency Retrieval.** We choose pretrained BGE-M3 and BM25 (Robertson et al., 2009) as baselines. BGE-M3 is a state-of-the-art embedding model that can perform accurate semantic retrieval for more than 100 languages. BM25 is a classical information retrieval method based on frequency and document length and is the main baseline in ReProver (Yang et al., 2024). For BGE-M3 baseline, we evaluate the pretrained model; For BM25, a BPE tokenizer with 30,000 vocabularies is trained on the topologically informalized Mathlib 4 dataset. For each ablative setting in experiments, we separately fine-tuned one retriever with the same recipe in Appendix A.8. Evaluation is conducted on the ProofNet (Azerbayev et al., 2023) and the Con-NF benchmark.

Results in Table 2 suggest the superiority of our method. Models fine-tuned on dependency retrieval dataset show landslide victory over baselines, exhibiting more than $10\times$ improvement of recall on ProofNet and $2\times$ on Con-NF. The huge performance gap between baselines focused on semantic similarity and our model indicates that dependency retrieval is a novel retrieval task, which relies more on logical dependency. For a more intuitive analysis, a case study can be found in Appendix A.6. Ablative results on topological informalization also demonstrate a consistent advantage over vanilla informalization, especially in OOD generalization (Con-NF), where relative improvements can reach $50\%$ on Recall@5 and Precision@5. Comparisons between formattings of formal objects indicate that incorporating informalizations in dependency embedding might introduce noise and degrade retrieval performance in in-distribution settings but improve OOD performance. We leave the exploration of this intriguing phenomenon for future work.

**Statement Autoformalization.** We evaluate a wide range of baselines, including in-context learning (Wu et al., 2022) using GPT-4o (OpenAI et al., 2024) and DeepSeek-V2.5 (DeepSeek-AI, 2024), and fine-tuning on MMA (Jiang et al., 2023a), PDA (Lu et al., 2024), and Lean-Workbook (Ying et al., 2024a). Since LLM API calling does not support beam search with $T = 0.0$, DeepSeek is evaluated using temperature decoding $T = 0.7$, and GPT-4o using version `gpt-4o-2024-08-06`

---

[6]For theorems, we only use their declarations since their code (except proofs) is identical with their declarations in semantics.

Table 2: Comparisons between our dependency retriever and baselines, and ablations of topological informalization. Cyan numbers in brackets show ablative improvements over vanilla informalization (**U**); Bold numbers emphasize the highest values in each benchmark; **Fmt** indicates the method to format a formal object into a string to embed, where **F** denotes using only formal declarations and **F+IF** means using both formal declarations and informalizations; **DR** represents dense retrieval; **Dataset** indicates the training dataset, where **P** means directly using pretrained model, **U** represents unstructurally informalized dataset, and **T** represents topologically informalized dataset; **R@$k$** and **P@$k$** denote the recall and precision of top-$k$ retrievals, respectively.

| Bench | Fmt | Method | Dataset | R@5↑ | R@10↑ | R@100↑ | P@5↑ | P@10↑ | P@100↑ |
|---|---|---|---|---|---|---|---|---|---|
| **ProofNet** | F | BM25 | T | 0.16% | 0.16% | 1.00% | 0.11% | 0.05% | 0.03% |
| | | DR | P | 1.93% | 2.13% | 7.14% | 1.02% | 0.61% | 0.24% |
| | | | U | 33.74% | 40.31% | 65.30% | 21.55% | 13.61% | 2.22% |
| | | | T | **35.52%** (1.79%) | **43.63%** (3.32%) | **67.71%** (2.42%) | **22.89%** (1.34%) | **14.57%** (0.96%) | 2.25% (0.03%) |
| | F+IF | BM25 | T | 0.00% | 0.11% | 0.29% | 0.00% | 0.05% | 0.01% |
| | | DR | P | 0.41% | 0.98% | 5.46% | 0.32% | 0.40% | 0.20% |
| | | | U | 28.66% | 34.57% | 63.55% | 18.18% | 11.28% | 2.16% |
| | | | T | 32.47% (3.81%) | 40.35% (5.78%) | 67.33% (3.78%) | 20.32% (2.14%) | 12.81% (1.52%) | **2.26%** (0.11%) |
| **Con-NF** | F | BM25 | T | 4.41% | 7.31% | 31.13% | 2.37% | 2.23% | 1.06% |
| | | DR | P | 5.66% | 9.10% | 34.50% | 3.73% | 3.02% | 1.15% |
| | | | U | 15.28% | 20.31% | 72.70% | 7.95% | 5.47% | 2.39% |
| | | | T | 24.32% (9.04%) | **37.44%** (17.13%) | **88.86%** (16.16%) | 14.05% (6.10%) | 11.29% (5.82%) | 3.19% (0.80%) |
| | F+IF | BM25 | T | 9.86% | 14.95% | 34.50% | 6.95% | 5.28% | 1.26% |
| | | DR | P | 13.84% | 19.19% | 44.16% | 9.51% | 6.72% | 1.59% |
| | | | U | 17.34% | 23.10% | 84.25% | 10.39% | 7.29% | 3.05% |
| | | | T | **27.91%** (10.57%) | 37.00% (13.90%) | 86.43% (2.18%) | **17.57%** (7.18%) | **11.99%** (4.69%) | **3.21%** (0.16%) |

and default hyperparameters. For both, we set repeat count $t = 8$ (retry if fail to extract formal statements from model outputs) and use 3-shot demonstrations. Notably, ProofNet participates in the data synthesis process of Lean-Workbook. But we still include it as a strong baseline. For fairness, all fine-tuning methods use InternLM2-Math-Base-7B (Ying et al., 2024b)[7] as base model and training recipe in Appendix A.8. We also report the performance of RAutoformalizer without retrieval module (RA -R) and given ground-truth dependencies (RA +R). Both are fine-tuned respectively on correspondingly constructed datasets. For ProofNet, additional objects defined beyond Mathlib 4 are retrieved in priority. For reproducibility, all fine-tuning methods are evaluated using beam search with temperature $T = 0.0$, generation number $n = 8$, and beam size $b = 8$.

We use BEq (introduced in Section 3.2) to evaluate the equivalence between model predictions and ground-truth formal statements. We define BEq@$k$ as the portion of samples where predicted statements are BEq to ground-truths at least once in $k$ attempts:

$$\text{BEq@}k = \frac{1}{N} \sum_{i=1}^{N} \max_{j \in \{1,\dots,k\}} \mathbb{I}_{\hat{\boldsymbol{s}}_{i,k} \sim_B \boldsymbol{s}_i} \tag{7}$$

where $N$ is the number of samples; $k$ is the number of attempts; $\mathbb{I}$ is the indicator function, and $\hat{\boldsymbol{s}}_{i,k}$ is the $j$-th generation attempt for the $i$-th sample. Similarly, Typecheck@$k$ is defined as the portion of samples where model predictions pass Lean typecheck at least once in $k$ attempts.

$$\text{Typecheck@}k = \frac{1}{N} \sum_{i=1}^{N} \max_{j \in \{1,\dots,k\}} \mathbb{I}_{\text{LeanTypecheck}(\hat{\boldsymbol{s}}_{i,k})} \tag{8}$$

We report BEq@1, BEq@8, Typecheck@1, and Typecheck@8 for a more thorough evaluation.

Table 3 shows the great superiority of RAutoformalizer over baselines. On in-distribution ProofNet benchmark, the non-retrieval ablative model already surpasses all baseline methods, including Lean-Workbook (Ying et al., 2024a) (by 6.69% in BEq@8), showing the high quality of our topological informalizations. RAutoformalizer further improves 1.60%. The ideal model reaches 23.26% BEq@1 and 31.28% BEq@8, exhibiting the potential of dependency retrieval.

On OOD Con-NF, without retrieval, all methods, including large-scale-pretrained GPT-4o and DeepSeek-V2.5, result in poor performance. The non-retrieval ablative model still shows the highest

---

[7]Another group of experiments fine-tuned on DeepSeek-Math-Base-7B can be found in Appendix A.3.

Table 3: Comparisons and ablations of retrieval-augment. Cyan numbers in brackets show ablative improvements over bare autoformalizer ("RA -R"); Bold numbers emphasize the highest values excluding oracle ("RA +R") results; **BEq@**$k$ indicates the portion of samples where predictions are equivalent to ground-truths under BEq at least once in $k$ attempts, defined in Eq. 7; **Typecheck@**$k$ indicates the portion of samples where predictions pass typecheck at least once in $k$ attempts, defined in Eq. 9; **ICL (D)** and **ICL(4o)** represents in-context learning using Deepseek-V2.5 and GPT-4o, respectively; **MMA**, **MMA (Lean)**, **PDA**, and **LW** represent fine-tuning on MMA, MMA's Lean subset, PDA, and Lean-workbook, respectively; **RA** is the main method; **RA -R** is the ablation removing dependency retrieval; **RA +R** is the ablation using oracle dependencies.

| Method | ProofNet | | | | Con-NF | | | |
|---|---|---|---|---|---|---|---|---|
| | Typecheck@1↑ | BEq@1↑ | Typecheck@8↑ | BEq@8↑ | Typecheck@1↑ | BEq@1↑ | Typecheck@8↑ | BEq@8↑ |
| ICL (D) | 40.37% | 9.89% | 51.07% | 10.96% | 9.37% | 2.81% | 16.23% | 4.27% |
| ICL (4o) | 43.58% | 7.22% | 66.31% | 12.83% | 9.78% | 1.46% | 20.71% | 4.16% |
| MMA | 12.57% | 1.87% | 22.99% | 2.94% | 3.64% | 1.98% | 8.74% | 4.37% |
| MMA (L) | 10.96% | 2.14% | 23.53% | 2.67% | 3.33% | 1.77% | 8.01% | 4.58% |
| PDA | 14.71% | 0.27% | 24.33% | 2.14% | 4.37% | 1.04% | 10.61% | 3.64% |
| LW | 44.92% | 8.56% | 49.20% | 9.89% | **28.10%** | 0.94% | **37.67%** | 1.04% |
| **RA -R** | 52.14% | 11.50% | 71.39% | 16.58% | 8.12% | 3.02% | 11.97% | 4.58% |
| **RA** | **57.22%** (5.08) | **12.30%** (0.80) | **77.27%** (5.88) | **18.18%** (1.60) | 20.50% (12.38) | **11.45%** (8.43) | 28.62% (16.65) | **16.86%** (12.28) |
| **RA +R** | 72.99% (20.86) | 23.26% (11.76) | 80.48% (9.09) | 31.28% (14.71) | 60.46% (52.34) | 44.85% (41.83) | 72.11% (60.15) | 55.36% (50.78) |

BEq@1 and BEq@8 among them. With retrieval-augment, RAutoformalizer has $3\times$ improvement on BEq@1 and BEq@8, and the oracle model exhibits over $10\times$ potential for improvement. This significant gap shows the necessity of dependency retrieval and draws community attention to OOD settings. A more detailed ablative study can be found in Appendix A.2.

## 5 CONCLUSION

We have presented a thorough rethink on existing statement autoformalization paradigms, identifying and addressing two critical problems: absence of universal human-aligned evaluation metric and agnosia of contextual information. For the first, we propose **BEq** (*Bidirectional Extended Definitional Equivalence*), a faithful, effective and universal neural-symbolic approach to determine the equivalence between formal statements. For the second, we propose a new task, *Dependency Retrieval*, finding dependent formal objects from math libraries, and a new paradigm, **RAutoformalizer** (*Retriever-augmented Autoformalizer*), enhancing statement autoformalization with dependency retrieval. We also propose to parse dependencies and topologically informalize formal objects to synthesize high-quality data. For more comprehensive evaluation, we extend ProofNet benchmark for dependency retrieval and construct a novel research-level OOD benchmark, Con-NF.

## 6 LIMITATION AND BROADER IMPACTS

**Limitations of BEq.** As an equivalence metric between formal statements, the accuracy of BEq depends on the quality of the ground-truth formal statements of the benchmarks. Therefore, BEq is not suitable for benchmarks with low-quality ground-truths or those lacking formal ground-truths.

Moreover, human opinions on equivalence are diverse. Therefore, carefully designing the limitation of transformation primitives $\mathcal{R}$ (available tactics) and the approximation of transformation function $T$ (the LLM) is crucial, for which extensive experiments are conducted in Appendix A.1. For the more detailed case study of BEq, please refer to Appendix A.4. We sincerely invite community efforts to delve into refining BEq and set a domain standard to facilitate subsequent research.

**Limitations of RAutoformalizer** For retrieval-augment generation, high-ranking retrievals mainly impact its performance (Cuconasu et al., 2024). Although RAutoformalizer surpasses all baselines by a significant margin, the experiment of oracle retrieval (RA +R) exhibits large room to improve the retriever. This project focuses on setting a basic working baseline for dependency retrieval and leaves sophisticated upgrades such as multi-vector embeddings (Khattab & Zaharia, 2020), re-ranking (Zhuang et al., 2022) and query augmentation (Gao et al., 2024) for future work.

**Broader Impacts.** We hope the idea of bidirectionally "convertible" under restricted transformations can inspire more areas, such as neural-symbolic, formal verification, and general reasoning. For example, faithful automated evaluation in other symbolic generative tasks. Furthermore, researchers can also extend RAutoformalizer to broader neural-symbolic tasks such as the autoformalization of specifications, proof, and even theories.

## 7 REPRODUCIBILITY STATEMENT

It aims to contribute to the field of statement autoformalization by proposing a faithful equivalence metric, a research-level benchmark, and a new paradigm for mitigating agnosia of context and enhancing OOD generalization. We fully understand the importance of reproducibility in scientific research and therefore, details of datasets, models, and experiments are summarized as follows:

- Implementation details of BEq in Section 3.2;
- Experiment settings and baselines for BEq in Section 3.3;
- Training dataset for dependency retriever and RAutoformalizer in Section 4.1, and string formatting details in Appendix A.8;
- Construction and composition of the Con-NF benchmark in Section 4.1;
- For dependency retriever, implementation details and experiment setting in Section 4.2, and detailed training recipe in Appendix A.8;
- For RAutoformalizer, implementation details, experiment setting, and evaluation metric in Section 4.2, and detailed training recipe in Appendix A.8;
- All dependent open-source libraries, along with their repository URLs and versions in Appendix A.9.

Evaluation results are uploaded as supplementary materials. Code, data, and model checkpoints are released at `https://github.com/Purewhite2019/rethinking_autoformalization`.

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

# A  APPENDIX

## A.1  COMPARATIVE EXPERIMENTS ON HYPERPARAMETERS OF BEQ

Extensive experiments are conducted to evaluate the influence of different engineering choices, as shown in Table 4. Experiment dimensions include the restrictions of transformation primitives, choices between BEq and only *Unidirectional Definitional Implication*, number of attempts to generate transformations, and sampling strategy.

As for the restrictions, **Basic** denotes only {`exact`, `exact?`, `have`} are allowed, **Normal** additionally includes {`apply`, `cases'`, `constructor`, `ext`, `intro`, `intros`, `rw`, `use`}, **Advanced** additionally allows more powerful tactics {`assumption`, `by_cases`, `by_contra`, `change`, `choose`, `convert`, `exfalso`, `left`, `nth_rw`, `obtain`, `rcases`, `refine`, `rfl`, `right`, `rintro`, `specialize`, `triv`}, and **All** denotes all tactics are allowed. Experiment results show that **Basic** setting is enough for most cases and **Normal** setting shows superior performance, while **Advanced** and **All** may lead to false positives.

Comparison between "Bidirectional" and "Unidirectional" shows landslide advantage of "Bidirectional". Experiments of $K$ show that symbolic heuristic `exact?` is able to handle most cases, but the incorporation of large language models can solve more cases. It also reveals that our current implementation, few-shot prompting LLM, is not capable of handling more difficult cases. Failure case analysis is done in Appendix A.4. The sampling strategy does not have much influence, so we use beam-search with temperature $T = 0$ in the main experiments.

## A.2  MORE RESULTS ON RAUTOFORMALIZER

### A.2.1  COMPREHENSIVE HUMAN EVALUATION OF BEQ

Table 3 compares the autoformalization performance of RA with other baselines on ProofNet and OOD Con-NF using two automated metrics: Typecheck and BEq. Because the robustness of BEq itself is limited as discussed above, the significance of the table results is compromised unless human evaluations are provided.

To more reliably evaluate BEq and RAutoformalizer, for each experiment on each benchmark, about 100 model predictions that pass the typecheck are sampled for human evaluation. To reduce the variance, we perform stratified sampling in 3 groups: 1) both directions of UDI (Unidirectional Definitional Implication) fail; 2) one single directional UDI succeeds; 3) both directions of UDI succeed (BEq). The results are shown in Table 5.

Results on ProofNet benchmark are consistent with Table 1. Moreover, BEq demonstrates nearly perfect accuracy on Con-NF. Therefore, BEq is robust as an automated evaluation metric for autoformalization tasks.

### A.2.2  HUMAN-RECTIFIED RESULTS

According to the human evaluation results in Appendix A.2.1, we can estimate the gold accuracy of our methods as

$$\text{Human@1} = \text{Typecheck@1} \times \text{HumanAcc}|_{\text{Typecheck}} \tag{9}$$

where Typecheck@1 is the portion of samples where predictions pass typecheck in one attempt; $\text{HumanAcc}|_{\text{Typecheck}}$ is the human evaluated model accuracy among sampled typechecked predictions in Appendix A.2.1. Results are shown in Table 6, which demonstrates clear ablative improvement among **RA -R**, **RA** and **RA +R** on the estimated goal accuracies.

### A.2.3  TYPECHECK ERROR DISTRIBUTION

To quantitatively delve into the underlying mechanics of the ablative improvement brought by RAutoformalizer, for each experiment, we count all Lean errors in samples that fail to typecheck and classify them into two sources: "Hallucination" (error caused by hallucination of identifiers) and "Others" (all other errors). The results are in Table 7, which show retrieval-augment can reduce both types of errors, especially Hallucination errors.

The detailed error taxonomy is as follows:

- `function expected`: Others
- `invalid field notation`: Hallucination
- `type expected, got`: Others
- `unknown constant`: Hallucination

Table 4: Comparative experiments of the proposed equivalence metric on the human-annotated equivalence benchmark. Green-backgrounded numbers are those reported in Table 1; Red-backgrounded numbers highlight false positives, which we're trying our best to avoid. **Restriction** represents the allowed transformation primitives.; **Bidirectional** indicates to determine equivalence by BEq; **Unidirectional** indicates to determine equivalence by *Unidirectional Definitional Implication*; **K** denotes the number of attempts to generate transformations; **T=0.0** means beam-search with temperature $T = 0$; **T=0.7** means temperature sampling with $T = 0.7$; **FP** denotes the number of false positives.

| Restriction | Direction | K | T=0.0 | | | | T=0.7 | | | |
|---|---|---|---|---|---|---|---|---|---|---|
| | | | FP | Precision | Recall | Accuracy | FP | Precision | Recall | Accuracy |
| Basic | Bidirectional | 0 | 0 | 100.00% | 67.14% | 88.50% | 0 | 100.00% | 67.14% | 88.50% |
| | | 1 | 0 | 100.00% | 70.00% | 89.50% | 0 | 100.00% | 70.00% | 89.50% |
| | | 2 | 0 | 100.00% | 70.00% | 89.50% | 0 | 100.00% | 70.00% | 89.50% |
| | | 4 | 0 | 100.00% | 70.00% | 89.50% | 0 | 100.00% | 70.00% | 89.50% |
| | | 8 | 0 | 100.00% | 70.00% | 89.50% | 0 | 100.00% | 70.00% | 89.50% |
| | | 16 | 0 | 100.00% | 71.43% | 90.00% | 0 | 100.00% | 71.43% | 90.00% |
| | Unidirectional | 0 | 16 | 75.00% | 68.57% | 81.00% | 16 | 75.00% | 68.57% | 81.00% |
| | | 1 | 18 | 73.91% | 72.86% | 81.50% | 18 | 74.29% | 74.29% | 82.00% |
| | | 2 | 18 | 74.29% | 74.29% | 82.00% | 19 | 73.61% | 75.71% | 82.00% |
| | | 4 | 19 | 73.61% | 75.71% | 82.00% | 19 | 74.32% | 78.57% | 83.00% |
| | | 8 | 19 | 74.32% | 78.57% | 83.00% | 21 | 72.37% | 78.57% | 82.00% |
| | | 16 | 23 | 71.25% | 81.43% | 82.00% | 23 | 70.89% | 80.00% | 81.50% |
| Normal | Bidirectional | 0 | 0 | 100.00% | 67.14% | 88.50% | 0 | 100.00% | 67.14% | 88.50% |
| | | 1 | 0 | 100.00% | 71.43% | 90.00% | 0 | 100.00% | 70.00% | 89.50% |
| | | 2 | 0 | 100.00% | 71.43% | 90.00% | 0 | 100.00% | 70.00% | 89.50% |
| | | 4 | 0 | 100.00% | 71.43% | 90.00% | 0 | 100.00% | 71.43% | 90.00% |
| | | 8 | 0 | 100.00% | 71.43% | 90.00% | 0 | 100.00% | 71.43% | 90.00% |
| | | 16 | **0** | **100.00%** | **72.86%** | **90.50%** | 0 | 100.00% | 72.86% | 90.50% |
| | Unidirectional | 0 | 16 | 75.00% | 68.57% | 81.00% | 16 | 75.00% | 68.57% | 81.00% |
| | | 1 | 17 | 74.63% | 71.43% | 81.50% | 18 | 74.29% | 74.29% | 82.00% |
| | | 2 | 18 | 74.29% | 74.29% | 82.00% | 18 | 74.29% | 74.29% | 82.00% |
| | | 4 | 19 | 73.97% | 77.14% | 82.50% | 18 | 74.29% | 74.29% | 82.00% |
| | | 8 | 19 | 74.32% | 78.57% | 83.00% | 19 | 74.32% | 78.57% | 83.00% |
| | | 16 | 22 | 71.79% | 80.00% | 82.00% | 20 | 74.03% | 81.43% | 83.50% |
| Advanced | Bidirectional | 0 | 0 | 100.00% | 67.14% | 88.50% | 0 | 100.00% | 67.14% | 88.50% |
| | | 1 | 0 | 100.00% | 70.00% | 89.50% | 0 | 100.00% | 71.43% | 90.00% |
| | | 2 | 0 | 100.00% | 71.43% | 90.00% | 0 | 100.00% | 71.43% | 90.00% |
| | | 4 | 0 | 100.00% | 71.43% | 90.00% | 0 | 100.00% | 71.43% | 90.00% |
| | | 8 | 0 | 100.00% | 71.43% | 90.00% | 0 | 100.00% | 71.43% | 90.00% |
| | | 16 | 0 | 100.00% | 72.86% | 90.50% | 1 | 98.08% | 72.86% | 90.00% |
| | Unidirectional | 0 | 16 | 75.00% | 68.57% | 81.00% | 16 | 75.00% | 68.57% | 81.00% |
| | | 1 | 18 | 73.53% | 71.43% | 81.00% | 17 | 75.36% | 74.29% | 82.50% |
| | | 2 | 18 | 74.29% | 74.29% | 82.00% | 17 | 75.71% | 75.71% | 83.00% |
| | | 4 | 18 | 75.00% | 77.14% | 83.00% | 19 | 74.32% | 78.57% | 83.00% |
| | | 8 | 22 | 71.43% | 78.57% | 81.50% | 22 | 71.79% | 80.00% | 82.00% |
| | | 16 | 24 | 70.37% | 81.43% | 81.50% | 26 | 68.29% | 80.00% | 80.00% |
| All | Bidirectional | 0 | 0 | 100.00% | 67.14% | 88.50% | 0 | 100.00% | 67.14% | 88.50% |
| | | 1 | 0 | 100.00% | 68.57% | 89.00% | 0 | 100.00% | 68.57% | 89.00% |
| | | 2 | 0 | 100.00% | 71.43% | 90.00% | 0 | 100.00% | 70.00% | 89.50% |
| | | 4 | 0 | 100.00% | 71.43% | 90.00% | 0 | 100.00% | 70.00% | 89.50% |
| | | 8 | 0 | 100.00% | 71.43% | 90.00% | 1 | 98.04% | 71.43% | 89.50% |
| | | 16 | 0 | 100.00% | 72.86% | 90.50% | 3 | 94.44% | 72.86% | 89.00% |
| | Unidirectional | 0 | 16 | 75.00% | 68.57% | 81.00% | 16 | 75.00% | 68.57% | 81.00% |
| | | 1 | 17 | 75.36% | 74.29% | 82.50% | 18 | 74.29% | 74.29% | 82.00% |
| | | 2 | 18 | 74.29% | 74.29% | 82.00% | 19 | 73.24% | 74.29% | 81.50% |
| | | 4 | 20 | 73.33% | 78.57% | 82.50% | 19 | 73.97% | 77.14% | 82.50% |
| | | 8 | 22 | 71.43% | 78.57% | 81.50% | 20 | 72.97% | 77.14% | 82.00% |
| | | 16 | 26 | 68.67% | 81.43% | 80.50% | 25 | 68.35% | 77.14% | 79.50% |

- `failed to synthesize instance`: Others
- `application type mismatch`: Others
- `unknown identifier`: Hallucination
- `invalid pattern, constructor or constant marked with :` Others
- `invalid pattern variable, must be atomic`: Others
- `unexpected end of input`: Others
- `unexpected token`: Others
- `invalid coercion notation, expected type is not known`: Others
- `cannot coerce to function`: Others

Table 5: Human evaluation results. **RA** is the main method; **RA -R** is the ablation removing dependency retrieval; **RA +R** is the ablation using oracle dependencies; **TP**, **TN**, **FP**, **FN** are the number of true-positives, true-negatives, false-positives and false-negatives of BEq, respectively.

| Benchmark | Method | BEq | | | | | | |
|---|---|---|---|---|---|---|---|---|
| | | TP | TN | FP | FN | Precision | Recall | Accuracy |
| **ProofNet** | **RA -R** | 22 | 70 | 0 | 9 | 100.00% | 70.97% | 91.09% |
| | **RA** | 22 | 67 | 0 | 11 | 100.00% | 66.67% | 89.00% |
| | **RA +R** | 32 | 57 | 0 | 12 | 100.00% | 72.73% | 88.12% |
| **Con-NF** | **RA -R** | 29 | 49 | 0 | 0 | 100.00% | 100.00% | 100.00% |
| | **RA** | 55 | 44 | 0 | 1 | 100.00% | 98.21% | 99.00% |
| | **RA +R** | 74 | 23 | 0 | 3 | 100.00% | 96.10% | 97.00% |

Table 6: Human-rectified results centering in RAutoformalizer ablative experiments. **RA** is the main method; **RA -R** is the ablation removing dependency retrieval; **RA +R** is the ablation using oracle dependencies; **BEq@1** indicates the portion of samples where predictions are equivalent to ground-truths under BEq in one attempt, defined in Eq. 7; **Typecheck@1** indicates the portion of samples where predictions pass typecheck in one attempt, defined in Eq. 9; **Human@1** indicates the estimated portion of samples where model predictions pass Human evaluation.

| Method | ProofNet | | | Con-NF | | |
|---|---|---|---|---|---|---|
| | Typecheck@1 | BEq@1 | Human@1 | Typecheck@1 | BEq@1 | Human@1 |
| **RA-R** | 52.14% | 11.50% | 16.00% | 8.12% | 3.02% | 3.02% |
| **RA** | 57.22% | 12.30% | 18.88% | 20.50% | 11.45% | 11.48% |
| **RA+R** | 72.99% | 23.26% | 31.80% | 60.46% | 44.85% | 46.55% |

- `typeclass instance problem is stuck, it is often due to metavariables`: Others
- `type mismatch`: Others
- `invalid {...} notation, expected type is not known`: Hallucination
- `stuck at solving universe constraint`: Others
- `invalid binder annotation, type is not a class instance`: Hallucination
- `invalid parametric local instance, parameter with type`: Others
- `invalid constructor ⟨...⟩, expected type must be an inductive type`: Hallucination
- `overloaded, errors` : Others
- `expected token`: Others
- `ambiguous, possible interpretations` : Others
- `don't know how to synthesize placeholder`: Others
- `invalid field, the environment does not contain`: Hallucination
- `invalid {...} notation, expected type is not of the form (C ...)`: Others
- `invalid dotted identifier notation, expected type is not of the form (... → C ...) where C is a constant`: Others
- `unexpected identifier`: Others
- `(deterministic) timeout at 'whnf maximum number of heartbeats (200000) has been reached (use 'set_option maxHeartbeats <num>' to set the limit)`: Others
- `failed to synthesize`: Others
- `failed to prove index is valid, possible solutions::` Others
- `cannot coerce to sort`: Others
- `invalid argument name`: Hallucination
- `don't know how to synthesize implicit argument`: Others
- `invalid projection`: Hallucination

Table 7: Distribution of typecheck errors in RAutoformalizer ablative experiments. **RA** is the main method; **RA -R** is the ablation removing dependency retrieval; **RA +R** is the ablation using oracle dependencies; **Hallucination** denotes the number of errors caused by hallucination, and **Others** denotes the number of other errors. Cyan numbers highlights the percentage of errors reduced relative to RA -R.

| Method | ProofNet | | Con-NF | |
|---|---|---|---|---|
| | **Hallucination** | **Others** | **Hallucination** | **Others** |
| **RA -R** | 434 | 1790 | 8902 | 14842 |
| **RA** | 320 (-26.27%) | 1500 (-16.20%) | 5217 (-41.40%) | 13386 (-9.81%) |
| **RA +R** | 65 (-85.02%) | 1173 (-34.47%) | 1134 (-87.26%) | 5882 (-60.37%) |

- `elaboration function has not been implemented`: Others
- `failed to infer binder type`: Others
- `invalid occurrence`: Others
- `invalid universe level`: Others
- `expected no space before`: Others
- `tactic failed`: Others
- `invalid constructor`: Others
- `missing end of character literal`: Others
- `unused universe parameter`: Others
- `unknown tactic`: Hallucination
- `unsolved goals`: Others
- `(kernel) declaration has metavariables`: Others
- `invalid use of field notation with '@' modifier`: Others
- `invalid {...} notation, structure type expected`: Others
- `unexpected syntax`: Others
- `expected ';' or line break`: Others
- `invalid binder name`: Hallucination
- `not a field of structure`: Hallucination
- `too many explicit universe levels`: Others
- `type class instance expected`: Others
- `fields missing`: Hallucination
- `invalid use of explicit universe parameters`: Others
- `is not a structure`: Hallucination
- `don't know how to synthesize placeholder for argument`: Others
- `cannot coerce`: Others
- `unknown universe level`: Others
- `expected structure`: Others
- `has already been declared`: Others
- `simp made no progress`: Others
- `missing cases:`: Others
- `invalid dotted identifier notation, unknown identifier`: Hallucination
- `invalid 'import' command, it must be used in the beginning of the file`: Others
- `(↑) must have a function type, not`: Others
- `not a structure`: Others

Table 8: Experiment results of fine-tuning-based autoformalization methods reproduced on DeepSeek-Math-Base-7B. Cyan numbers in brackets show ablative improvements over bare autoformalizer ("RA -R"); Bold numbers emphasize the highest values excluding oracle ("RA +R") results; **BEq@**$k$ indicates the portion of samples where predictions are equivalent to ground truths under BEq at least once in $k$ attempts, defined in Eq. 7; **Typecheck@**$k$ indicates the portion of samples where predictions pass typecheck at least once in $k$ attempts, defined in Eq. 9; **MMA**, **MMA (Lean)**, **PDA**, and **LW** represent fine-tuning on MMA, MMA's Lean subset, PDA, and Lean-workbook, respectively; **RA** is the main method; **RA -R** is the ablation removing dependency retrieval; **RA +R** is the ablation using oracle dependencies.

| Method | ProofNet | | | | Con-NF | | | |
|---|---|---|---|---|---|---|---|---|
| | Typecheck@1↑ | BEq@1↑ | Typecheck@8↑ | BEq@8↑ | Typecheck@1↑ | BEq@1↑ | Typecheck@8↑ | BEq@8↑ |
| MMA | 15.78% | 1.87% | 31.02% | 5.08% | 3.23% | 1.66% | 7.28% | 4.06% |
| MMA (L) | 17.65% | 2.41% | 31.02% | 5.61% | 2.71% | 1.35% | 7.39% | 4.37% |
| PDA | 14.71% | 2.14% | 27.54% | 5.61% | 4.89% | 1.77% | 10.82% | 4.47% |
| LW | 36.10% | 8.56% | 53.74% | 10.16% | 4.89% | 1.98% | 11.13% | 2.08% |
| RA -R | 51.34% | 10.96% | 69.79% | 15.24% | 8.22% | 3.12% | 12.59% | 4.27% |
| RA | **59.36%** (8.02%) | **10.96%** (0.00%) | **75.94%** (6.15%) | **17.91%** (2.67%) | **17.59%** (9.37%) | **9.68%** (6.56%) | **25.49%** (12.90%) | **15.30%** (11.03%) |
| RA +R | 72.73% (13.37%) | 23.80% (12.83%) | 83.69% (7.75%) | 32.62% (14.71%) | 60.56% (42.98%) | 44.02% (34.34%) | 75.96% (50.47%) | 59.00% (43.70%) |

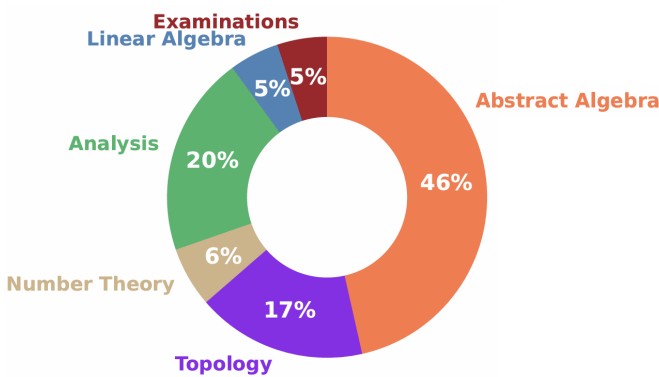

Figure 3: Disciplines distribution in the benchmark for human evaluation.

## A.3 EXPERIMENT RESULTS ON DEEPSEEK-MATH-BASE-7B

We also evaluate all fine-tuning-based methods using DeepSeek-Math-Base-7B (Shao et al., 2024) as the base model and the training recipe shown in Appendix A.8. The results are in Table 8, which demonstrate consistent (and even clearer) advantage of our methods over all baselines, and the ablative improvement of Dependency Retrieval.

## A.4 HUMAN EVALUATION FOR BEQ

**Human Equivalence Benchmark.** We use an early version of RAutoformalizer with oracle dependency (RA +R) and OpenAI o1-preview to predict formal statements for all samples in ProofNet (Azerbayev et al., 2023) benchmark. RAutoformalizer uses greedy decoding, while o1-preview uses temperature decoding with default hyperparameters from OpenAI. Generated statements are then filtered by typecheck and deduplicated by string matching. Then we uniformly sample 100 statement pairs from each model's generation, invite human experts from diverse backgrounds to label them as "equivalent" or "inequivalent", resulting in our *Human Equivalence Benchmark*. In total, 4 experts, one from formal verification and three from computer science participate in the labeling. They first separately evaluate the equivalence between formal statements, and discuss in round-table to reach an agreement for each sample. The distribution of disciplines in this benchmark is visualized in Figure 3

**Failure Case Analysis.** Our BEq reaches 100% precision, thus there are no false positives. For false negatives, we analyze them in detail and find roughly 2 error patterns:

- Semantic gaps between informal mathematics and formal verification. 9 out of 19 false negatives stem from it. Some subtle differences in informal mathematics may result in large differences between formalizations. As illustrated in Figure 4, formalization $P$ and $Q$ are identical in semantics, but they are formalized under different bases, one by subtype and the other by set. Another example is Figure 5, where model-generated proof fails due to a subtle but fatal difference in the underlying types.

- Transformation function failure. 10 out of 19 false negatives stem from it. Proving unidirectional definitional implication is a novel task, hence the prohibitive lack of supervised data makes it impossible to fine-tune a capable model. Our implementation utilizes a 5-shot prompted 20B model,

**Informal Statement** Show that there are infinitely many primes congruent to $-1$ modulo 6.

**Formalization P**
```
theorem sP :
    Infinite {p : Nat.Primes // p ≡ -1 [ZMOD 6]} :=
sorry
```
**Formalization Q**
```
theorem sQ :
  Set.Infinite {p : ℕ | Nat.Prime p ∧ p % 6 = 5} :=
sorry
```

Figure 4: Failure case of BEq: small semantic gap for natural language mathematics might be huge for formal verifier

**Informal Statement** Let $R$ be a ring in which $x^3 = x$ for every $x \in R$. Prove that $R$ is commutative.

**Formalization P**
```
theorem sP {R : Type*} [Ring R]
  (h : ∀ x : R, x ^ 3 = x) :
    CommRing R :=
sorry
```
**Formalization Q**
```
theorem sQ {R : Type u_1} [Ring R]
  (h : ∀ (x : R), x ^ 3 = x) (x : R) (y : R) :
    x * y = y * x :=
sorry
```
**Failed Proof**
```
have h_comm := exercise_4_2_5 h
have h_xy := h_comm.mul_comm x y
h_xy

---

type mismatch
  h_xy
has type
  @HMul.hMul R R R (@instHMul R NonUnitalNonAssocSemiring.toMul) x y = y * x : Prop
but is expected to have type
  @HMul.hMul R R R (@instHMul R NonUnitalNonAssocRing.toMul) x y = y * x : Prop
```

Figure 5: Failure case of BEq: imperceptible differences in type are intolerable in Lean.

which is relatively weak and fails to generate proper transformation for more complex scenarios, as illustrated in Figure 6 and Figure 7.

**Success Case Analysis.** Due to its symbolic nature, BEq can easily find fundamental differences between formalizations that are misleading for human expert. We demonstrate two examples in Figure 8 and Figure 9.

## A.5 VISUALIZATION OF BLEU DISTRIBUTION

The distribution of BLEU scores between formal statement pairs from the Human Equivalence Benchmark are visualized in Figure 10, along with the precision, recall, and accuracy curves w.r.t. different thresholds.

**Informal Statement** Prove that no group of order $pq$, where $p$ and $q$ are prime, is simple.

**Formalization P**
```
theorem sP {G : Type*} [Group G] [Fintype G] {p q : ℕ}
  (hp : Prime p) (hq : Prime q) (hG : card G = p*q) :
  IsSimpleGroup G ⟹ False :=
sorry
```
**Formalization Q**
```
theorem sQ
    (p q : ℕ)
    (hp : Nat.Prime p)
    (hq : Nat.Prime q)
    (G : Type _) [Group G] [Fintype G]
    (hG : Fintype.card G = p * q)
    : ¬ IsSimpleGroup G :=
sorry
```
**Equivalence Proofs**
$s_P \sim_B T(s_Q|s_P, \mathcal{R})$
```
have hpp : Prime p := by exact Nat.prime_iff.mp hp
have hqq : Prime q := by exact Nat.prime_iff.mp hq
exact sP hpp hqq hG
```
$s_Q \sim_B T(s_P|s_Q, \mathcal{R})$
```
have hpp : Nat.Prime p := by exact Nat.prime_iff.mpr hp
have hqq : Nat.Prime q := by exact Nat.prime_iff.mpr hq
exact sQ p q hpp hqq G hG
```

Figure 6: Failure case of BEq: transformation function fails to generate the transformation.

**Informal Statement** Assume that $f \colon \mathbb{R} \to \mathbb{R}$ satisfies $|f(t) - f(x)| \leq |t - x|^2$ for all $t, x$. Prove that $f$ is constant.

**Formalization P**
```
theorem sP {f : ℝ ↦ ℝ}
  (hf : ∀ x y, |f x − f y| ≤ |x − y| ^ 2) :
  ∃ c, f = λ x => c :=
sorry
```
**Formalization Q**
```
theorem sQ (f : ℝ ↦ ℝ )
  (h : ∀ (t x : ℝ ), |f t − f x| ≤ |t − x| ^ 2)
  (x : ℝ ) (y : ℝ ) : f x = f y :=
sorry
```
**Equivalence Proofs**
$s_P \sim_B T(s_Q|s_P, \mathcal{R})$
```
have hc := sQ f hf
use f 0
ext x
exact hc x 0
```
$s_Q \sim_B T(s_P|s_Q, \mathcal{R})$
```
have hc := sP h
cases' hc with c hc
have hx : f x = c := by exact congrFun hc x
have hy : f y = c := by exact congrFun hc y
rw [hx, hy]
```

Figure 7: Failure case of BEq: transformation function fails to generate the transformation.

**Informal Statement** Show that $\sin(\pi/12)$ is an algebraic number.

**Formalization P**
```
theorem sP : IsAlgebraic ℚ (sin (pi/12)) :=
  sorry
```
**Formalization Q**
```
theorem sQ : IsAlgebraic ℚ (Real.sin (Real.pi/12)) :=
  sorry
```

Figure 8: Success case of BEq: These two formalizations are not equivalent. Note that `pi` in Formalization P is an implicit argument of an arbitrary real number, instead of $\pi$.

**Informal Statement** Prove that $x^6 + 30x^5 - 15x^3 + 6x - 120$ is irreducible in $\mathbb{Z}[x]$.

**Formalization P**
```
theorem sP : Irreducible
  (X^6 + 30*X^5 - 15*X^3 + 6*X - 120 : Polynomial ℤ) :=
sorry
```
**Formalization Q**
```
theorem sQ
  (f : Polynomial ℤ := X^6 + 30*X^5 - 15*X^3 + 6*X - 120)
  : Irreducible f :=
sorry
```

Figure 9: Success case of BEq: These two formalizations are not equivalent. Note that `f : Polynomial ℤ :=X^6+30*X^5-15*X^3+6*X-120` in Formalization P means `f` is of type `Polynomial ℤ` with default parameter `X^6+30*X^5-15*X^3+6*X-120`, instead of `f=X^6+30*X^5-15*X^3+6*X-120`.

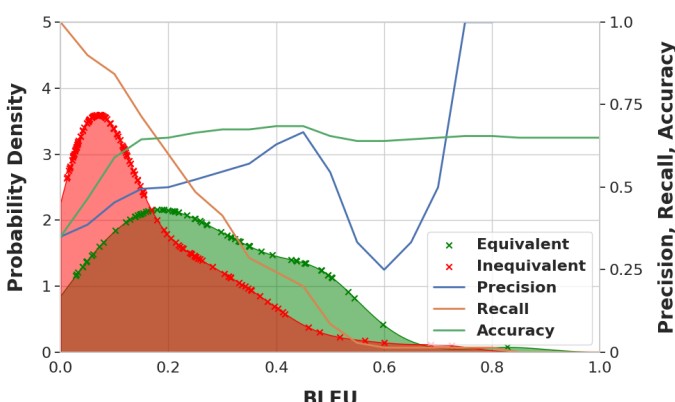

Figure 10: Distribution of BLEU in the benchmark and precision, recall, accuracy of different BLEU thresholds.

### A.6 CASE STUDY OF BM25 RETRIEVAL

Formally, BM25 (Robertson et al., 2009) can be defined as follows:

$$\text{BM25}(\boldsymbol{d}, \boldsymbol{q}) = \sum_{i=1}^{n} \text{IDF}(q_i, \boldsymbol{D}) \frac{(k_1 + 1)f(q_i, \boldsymbol{d})}{f(q_i, \boldsymbol{d}) + k_1(1 - b + b \cdot \frac{\text{Len}(\boldsymbol{d})}{\text{Mean}(\{\text{Len}(\boldsymbol{d}')|\boldsymbol{d}' \in \boldsymbol{D}\})})}$$

$$\text{IDF}(q_i, \boldsymbol{D}) = \log(\frac{N - |\{q_i \in \boldsymbol{d}|\boldsymbol{d} \in \boldsymbol{D}\} + 0.5}{|\{q_i \in \boldsymbol{d}|\boldsymbol{d} \in \boldsymbol{D}\}| + 0.5} + 1)$$

---

**Query**
Suppose that $f$ is holomorphic in an open set $\Omega$. Prove that if $\text{Re}(f)$ is constant, then $f$ is constant.
**Ground-truth Document**
Function 'Set' maps a given type to a proposition, which means that for each element of that type, it determines whether that element belongs to the set. A set a collection of elements of some type $\alpha$.
**Irrelevant Document 1**
If a function 'f' from a complex manifold 'M' to a complex normed space 'F' is holomorphic on a preconnected, compact, and open set 'U', and 'a' and 'b' are points in 'U', then 'f a = f b'.
**Irrelevant Document 2**
If a function 'f' from a topological space 'X' to a type 'Y' is locally constant, then for any point 'x' in 'X', there exists an open set 'U' containing 'x' such that 'f' is constant on 'U'.

---

Figure 11: Failure case of BM25: BM25 prefers semantic similarity to logical dependency.

where $\boldsymbol{q} = \{q_i\}_{i=1}^{n}$ is a query with $n$ tokens $q_1, \ldots, q_n$; $\boldsymbol{D} = \{\boldsymbol{d}_i\}_{i=1}^{N}$ is a document collection with $N$ documents $\boldsymbol{d}_i, \ldots; \boldsymbol{d}_N$, $k_1$ and $b$ are hyperparameters; $\text{IDF}(q_i, \boldsymbol{D})$ is the inverse document frequency of token $q_i$ in document $\boldsymbol{D}$.

As Figure 11 shows, BM25 prefers "semantic similarity" to "logical dependency" during retrieval. We focus on 3 keywords, holomorphic, set, and constant in the query. The query depends on the definition of "Set", but the frequencies of two keywords holomorphic and constant are 0 in the definition of "Set". Instead, the first irrelevant document shares similar frequency of set and holomorphic, while the second irrelevant one is similar in set and constant. Subsequently, both irrelevant documents have higher BM25 scores than the ground-truth.

## A.7 PROMPT TEMPLATES

### A.7.1 PROMPT TEMPLATE OF BEQ

```
Given two Lean 4 theorems, please prove 'thm_Q' with 'thm_P'.
You can only use the following tactics: {ALLOWED_TACTICS}
'thm_P' should be used at least once in the proof.
DO NOT add any extra explanation.
Here are some examples:

Input:
```
import Mathlib

open Topology Filter Real Complex TopologicalSpace Finset
open scoped BigOperators
noncomputable section

theorem thm_P : \not \exists (x : Rat), ( x ^ 2 = 12 ) :=
sorry

theorem thm_Q (q : Rat ) :q ^ 2 \neq 12 := by
```
Output:
```
exact (not_exists.mp thm_P) q
```

---

Input:
```
import Mathlib

open Fintype Subgroup Set Polynomial Ideal
open scoped BigOperators
noncomputable section
```

```
theorem thm_P {p q r : Nat} {G : Type*} [Group G]
  [Fintype G]   (hpqr : p < q \and q < r)
  (hpqr1 : p.Prime \and q.Prime \and r.Prime)(hG : card G = p*q*r) :
  Nonempty (Sylow p G) \or Nonempty (Sylow q G) \or Nonempty (Sylow r G) :=
sorry

theorem thm_Q {p : Nat } {q : Nat } {r : Nat } {G : Type u_1} [Group G] [Fintype G] (hp
    : Nat.Prime p) (hq : Nat.Prime q) (hr : Nat.Prime r) (hpq : p < q) (hqr : q < r)
    (hG : Fintype.card G = p * q * r) :Nonempty (Sylow p G) \or Nonempty (Sylow q G)
    \or Nonempty (Sylow r G) := by
```
Output:
```
exact thm_P (And.intro hpq hqr)  (And.intro hp (And.intro hq hr)) hG
```

---

Input:
```
import Mathlib

open Fintype Complex Polynomial LinearMap FiniteDimensional Module Module.End
open scoped BigOperators

theorem thm_P {F V : Type*} [AddCommGroup V] [Field F]
  [Module F V] (S T : End F V) :
  (S * T).Eigenvalues = (T * S).Eigenvalues :=
sorry

theorem thm_Q {K : Type v} {V : Type w} [Field K] [AddCommGroup V] [Module K V] (S :
    Module.End K V) (T : Module.End K V) :Module.End.Eigenvalues (S * T) =
    Module.End.Eigenvalues (T * S) := by
```
Output:
```
exact @thm_P K V _ _ _ S T
```

---

Input:
```
import Mathlib

open Function Fintype Subgroup Ideal Polynomial Submodule Zsqrtd
open scoped BigOperators
noncomputable section

theorem thm_P
    {p : Nat} {hp : Nat.Prime p} (h : \exists r : Nat, p = 2 ^ r + 1) :
    \exists (k : Nat), p = 2 ^ (2 ^ k) + 1 :=
sorry

theorem thm_Q {p : Nat } (hp : Nat.Prime p) (h : \exists (r : Nat ), p = 2 ^ r + 1)
    :\exists (k : Nat ), p = 2 ^ 2 ^ k + 1 := by
```
Output:
```
exact @thm_P p hp h
```
```

---

Input:
```
import Mathlib

open Fintype Set Real Ideal Polynomial
open scoped BigOperators
noncomputable section

theorem thm_P {G : Type*} [Group G]
  [Fintype G] (hG2 : Even (card G)) :
  \exists (a : G), a \neq 1 \and a = a\-1 :=
sorry

theorem thm_Q {G : Type*} [Group G] [Fintype G] (h : Fintype.card G % 2 = 0) :
    \exists a : G, a \neq 1 \and a = a\-1 := by
```
Output:
```
have hG : Even (card G) := by exact?
exact thm_P hG
```

---

According to the task description and examples, given the following two Lean 4
    theorems, please prove 'thm_Q' with 'thm_P'.

Input:
```
{THMS_TO_EVALUATE}
```
Output:

To apply this template, {ALLOWED_TACTICS} should be replaced to the list of allowed tactics and
{THMS_TO_EVALUATE} be replaced to the two statements to evaluate.

### A.7.2 PROMPT TEMPLATE OF LLM GRADER

**Backtranslation Template**
Given a Lean 4 theorem, please **briefly** and **consisely** explain it in natural
    language in one line.
Here are some examples:

Code:
```
theorem putnam_1964_b3
(f : Real \imp Real)
(hf : Continuous f \and \forall \alpha > 0, Tendsto (fun n : Nat \mapsto f (n *
    \alpha)) atTop (\nhds 0))
: (Tendsto f atTop (\nhds 0)) := sorry
```
Summarization: Suppose $f : \mathbb{R} \to \mathbb{R}$ is continuous and for every $\alpha > 0,\ \lim_{n\to\infty} f(n\alpha) = 0$.
    Prove that $\lim_{x\to\infty} f(x) = 0$.

---

Code:
```
theorem putnam_1968_b2
[Group G]
(hG : Finite G)
```

```
(A : Set G)
(hA : A.ncard > (Nat.card G : \Rat)/2)
: \forall g : G, \exists x \in A, \exists y \in A, g = x * y := by sorry
```
Summarization: Let $G$ be a finite group (with a multiplicative operation), and $A$ be a
    subset of $G$ that contains more than half of $G'$s elements. Prove that every
    element of $G$ can be expressed as the product of two elements of $A$.

---

Code:
```
theorem putnam_2022_a3
(p : Nat)
(hp : Nat.Prime p \and p > 5)
(f : Nat := {a : Nat \imp (ZMod p) | \forall n : Nat, a n \neq 0 \and a n * a (n + 2) =
    1 + a (n + 1)}.ncard)
: f \equiv 0 [MOD 5] \or f \equiv 2 [MOD 5] := sorry
```
Summarization: Let $p$ be a prime number greater than 5. Let $f(p)$ denote the number of
    infinite sequences $a_1, a_2, a_3, \ldots$ such that $a_n \in \{1, 2, \ldots, p-1\}$ and
    $a_n a_{n+2} \equiv 1 + a_{n+1} \pmod{p}$ for all $n \geq 1$. Prove that $f(p)$ is congruent to 0 or 2
    $\pmod{5}$.

Please **briefly** and **consisely** explain the following theorem in one line:
Code:
```
{THM_CODE}
```
Summarization:

To apply this template, $\{$THM$\_$CODE$\}$ should be replaced to the formal statement to informalize.

**Equivalence Determination Template**

Please check following two math problems is same or different? Please consider each
    statement in two problems, they are different if any statement is different.
    Please point out any differences you found. Please reply **same** or **different**
    in the final sentence with bold format.

Problem 1: {THM_1}

Problem 2: {THM_2}

To apply this template, $\{$THM$\_1\}$ and $\{$THM$\_1\}$ should be replaced to the informalizations of the two formal
statements to evaluate. Notably, when Majority Voting is adopted, it is recommended to randomize the order of
the two statements in multiple attempts.

### A.7.3 PROMPT TEMPLATE OF ICL AUTOFORMALIZATION

Please translate mathematical propositions into Lean 4 theorems. 'Mathlib' is the only
    allowed import.
DO NOT add any imports into the translation, and DO NOT try to prove the theorem, ONLY
    translate it.

Here are some examples:

Math Proposition:
'''
Suppose $f : \mathbb{R} \to \mathbb{R}$ is continuous and for every $\alpha > 0$, $\lim_{n \to \infty} f(n\alpha) = 0$. Prove that
    $\lim_{x \to \infty} f(x) = 0$.
'''
Lean Theorem:
```
theorem exercise
    (f : Real \implies Real)
```

```
        (hf : Continuous f \and \forall \alpha > 0, Tendsto (fun n : Nat \mapsto f (n *
            \alpha)) atTop (\nhds 0))
        : (Tendsto f atTop (\nhds 0)) :=
sorry
```

Math Proposition:
```
Let G be a finite group (with a multiplicative operation), and A be a subset of G
    that contains more than half of G's elements. Prove that every element of G can
    be expressed as the product of two elements of A.
```
Lean Theorem:
```
theorem exercise
    [Group G]
    (hG : Finite G)
    (A : Set G)
    (hA : A.ncard > (Nat.card G : Rat)/2)
    : \forall g : G, \exists x \in A, \exists y \in A, g = x * y :=
sorry
```

Math Proposition:
```
Let p be a prime number greater than 5. Let f(p) denote the number of infinite
    sequences a_1, a_2, a_3, ... such that a_n \in \{1, 2, ..., p - 1\} and a_n a_{n+2} \equiv 1 + a_{n+1} \pmod{p}
    for all n \geq 1. Prove that f(p) is congruent to 0 or 2 \pmod{5}.
```
Lean Theorem:
```
theorem exercise
    (p : Nat)
    (hp : Nat.Prime p \and p > 5)
    (f : Nat := {a : Nat \implies (ZMod p) | \forall n : Nat, a n \neq 0 \and a n * a
        (n + 2) = 1 + a (n + 1)}.ncard)
    : f \equiv 0 [MOD 5] \or f \equiv 2 [MOD 5] :=
sorry
```

Please translate the following proposition:
Math Proposition:
```
{INFORMAL_STMT}
```
Lean Theorem:

To apply this template, {INFORMAL_STMT} should be replaced to the informal statement to autoformalize.

**Equivalence Determination Template**

```
Please check following two math problems is same or different? Please consider each
    statement in two problems, they are different if any statement is different.
    Please point out any differences you found. Please reply **same** or **different**
    in the final sentence with bold format.

Problem 1: {THM_1}

Problem 2: {THM_2}
```

To apply this template, {THM_1} and {THM_1} should be replaced to the informalizations of the two formal statements to evaluate. Notably, when Majority Voting is adopted, it is recommended to randomize the order of the two statements in multiple attempts.

## A.8 FINE-TUNING DETAILS

**Dependency Retriever.** We fine-tune dependency retriever based on BGE-M3 (Chen et al., 2023) with FlagEmbedding library. Query string is identical to informalizations of theorems. The composition of document strings is as follows:

- F+IF: `Formal Declaration:decl\nInformal Explanation:if_stmt`
- F: `Formal Declaration:decl`

where `decl` and `if_stmt` represents formal declarations and informalizations, resepectively. Both are clipped to 1536 characters at most before composition.

We follow the default hyperparameters of FlagEmbedding, which are as follows:

- Learning Rate: $5 \times 10^{-6}$
- Warmup Ratio: 0.1
- Weight Decay: 0.01
- Precision: `fp16`
- Train Epochs: 6
- Gradient Accumulation Steps: 32
- Per Device Train Batch Size: 2
- Training Devices: 8
- Dataloader Drop Last: True
- Normalized: True
- Temperature: 0.02
- Query Max Length: 1024
- Passage Max Length: 1024
- Training Group Size: 4
- Hard Negative Size: 2
- Negatives Cross Device: False
- Query Instruction For Retrieval: None
- Inbatch Negative: False

**RAutoformalizer.** RAutoformalizer and all fine-tuning experiments are fine-tuned from InternLM2-Math-Base-7B (Ying et al., 2024b) using XTuner (Contributors, 2023) and the following hyperparameters:

- Max Sequence Length: 8192
- Variable-length Attention: True
- Batch size: 1
- Gradient Accumulation: 4
- Training Devices: 8
- Train Epochs: 1
- Optimizer: AdamW with learning rate $2\times10^{-5}$, $\boldsymbol{\beta} = (0.9, 0.999)$, weight decay 0, maximal gradient norm 1, warpup ratio 0.03 and `float16` mixed precision training.
- Learning Rate Scheduler: Warmup using LinearLR with start factor $10^{-5}$, then train using CosineAnnealingLR with $\eta_{\min} = 0.0$.

## A.9 OPEN-SOURCE LIBRARIES

For reproducibility, all relevant open-source projects are summarized in Table 9. Special thanks to the authors of these excellent projects.

Table 9: Versions of open-source projects used in this project.

| Name | Github Link | Version |
|---|---|---|
| FlagEmbedding | https://github.com/FlagOpen/FlagEmbedding | 76080ab83216d6d4156a597b220764a5bda45d92 |
| Xtuner | https://github.com/InternLM/xtuner | 0.1.23 |
| Lean 4 | https://github.com/leanprover/lean4 | 4.7.0-rc2 |
| Mathlib 4 | https://github.com/leanprover-community/mathlib4 | 59fdb6b04d7d16825a54483d550d9572ff473abf |
| REPL | https://github.com/leanprover-community/repl | 2ab7948163863ee222891653ac98941fe4f20e87 |
| Doc-Gen 4 | https://github.com/leanprover/doc-gen4 | 780bbec107cba79d18ec55ac2be3907a77f27f98 |
| ProofNet-lean4 | https://github.com/rahul3613/ProofNet-lean4 | 60efffb605ee07bf723db4fb8058129a7c8a89bb |
| LeanDojo | https://github.com/lean-dojo/LeanDojo | 78cee9d37aa32e70cdd6119c4af70ae551b8b713 |
| Con-NF | https://github.com/leanprover-community/con-nf | 16041ae6ea8b9a2ca79952afc7b927ccea18697b |

Table 10: Experiment results of augmenting in-context learning methods by dependency retrieval. Bold numbers emphasize the highest values excluding oracle results; **BEq@**$k$ indicates the portion of samples where predictions are equivalent to ground truths under BEq at least once in $k$ attempts, defined in Eq. 7; **T@**$k$ indicates the portion of samples where predictions pass typecheck at least once in $k$ attempts, defined in Eq. 9; **ICL** represents in-context learning using 3-shot demonstrations; **ICL+RA** represents in-context learning using 3-shot demonstrations, augmented by dependency retriever trained in Sec. 4.2; **ICL+RA** represents in-context learning using 3-shot demonstrations, augmented with ground-truth dependencies; **D-2.5** denotes using Deepseek-V2.5.

| Benchmark | | ProofNet | | | | Con-NF | | | |
|---|---|---|---|---|---|---|---|---|---|
| LLM | Method | T@1 | Beq@1 | T@8 | Beq@8 | T@1 | Beq@1 | T@8 | Beq@8 |
| | ICL | 43.58% | **7.22%** | 66.31% | 12.83% | 9.78% | 1.46% | 20.71% | 4.16% |
| GPT-4o | ICL+RA | **46.52%** | 6.95% | **77.01%** | **13.37%** | **22.79%** | **6.66%** | **50.57%** | **12.59%** |
| | ICL+RA (+R) | 58.56% | 17.38% | 81.28% | 29.14% | 54.84% | 38.40% | 75.75% | 54.11% |
| | ICL | 40.37% | **9.89%** | 51.07% | 10.96% | 9.37% | **2.81%** | 16.23% | **4.27%** |
| D-2.5 | ICL+RA | **43.32%** | 6.42% | **58.82%** | 10.96% | 9.37% | 1.87% | 15.19% | 3.12% |
| | ICL+RA (+R) | 61.50% | 17.91% | 72.99% | 20.32% | 48.18% | 32.36% | 62.02% | 41.94% |

## A.10 EXPERIMENT OF AUGMENTING ICL METHODS BY DEPENDENCY RETRIEVAL

The performance of augmenting **ICL** (in-context learning) methods with Dependency-retrieval-augmentation is shown in Table 10.

For GPT-4o, the results meet our expectations: **RA** consistently improves all metrics on all benchmarks (except BEq@1 on ProofNet), and **RA(+R)** shows the potential of dependency retrieval.

However, for Deepseek-V2.5, **RA** doesn't work well. We hypothesize this might be because the instruction-following and long-context capabilities of Deepseek-V2.5 are limited, thus the noise in retrieved dependencies degrades autoformalization. But **RA (+R)** shows significantly better performance than expected.

