# OpenReview forum: "Rethinking and Improving Autoformalization: Towards a Faithful Metric and a Dependency Retrieval-based Approach"
_ICLR.cc/2025/Conference — ICLR 2025 Spotlight_

### Official Review · Reviewer_6DKY · 2024-11-01

**Soundness:** 3
**Presentation:** 3
**Contribution:** 3
**Rating:** 8
**Confidence:** 5

**Summary:**

This paper improves the statement autoformalization by identifying and solving two key problems. First, they proposed BEq, a better LLM-based way to check if autoformalizations are correct by combining symbolic rules, and the human evaluation proves it's promising. Second, they created RAutoformalizer, a RAG approach, which looks up relevant dependent objects for auto-formalization. Also, authors annotated an OOD benchmark for further evaluation. The experiments show their methods are better than baselines.

**Strengths:**

1) This paper propose a new LLM-based evaluator BEq for statement auto-formalization, which make it more flexible as human evalution.
2) Authors designed a RAG-based auto-formalization framework by Dependency Retrieval
3) Authors annotate a new benchmark to evaluate OOD performance of their method.
4) Experiments show their fine-tuned retriever and whole RAG system more effective than baselines.

**Weaknesses:**

In general, this is a well-structured and written papers. But I still have some concerns about each components of contributions.

### Introduction Narrative:
1) In the Line 90-99, it would be more clear if authors can first state why they propose **dependency retrieval**. Why this new task is essential in the context of current methods.
2) In Line 65-75, it would be more helpful for authors to provide more contexts like citations to introduce statements, i.e.,
    - The discussion of β-reduction and type checking limitations (Lines 65-75) needs citations or examples demonstrating these issues
    - The statement about type checking being "too weak" requires empirical evidence or references to prior work.
3) Authors state some disadvantages of "LLM graders". Could authors provide more citations stating this?
    - The paper critiques LLM graders as "non-determinant and highly dependent on prompts" but doesn't sufficiently distinguish how BEq (which also uses LLMs) overcomes these limitations.
    - The few-shot prompting used in BEq seems to face similar prompt-design challenges that the authors criticize.
    - While Table 1 shows improved performance, more analysis is needed to demonstrate that BEq fundamentally addresses the cited limitations of LLM graders raised in the introduction.

### BEq Evaluator:
1) How few-shot prompting design in BEq pipeline can release human designed efforts of previous works as mentioned in the line 66-68 (Introduction)?
2) It seems that the BEq is implemented by a specific math LLM, InternLM2-Math-Plus-20B. Choice of InternLM2-Math-Plus-20B needs justification:
   - Is its math pretraining superior to alternatives?
   - Table 1 shows DeepSeek V2.5 outperforms InternLM2-Math-Plus-20B in majority voting
   - Why not implement BEq with DeepSeek V2.5? Is model size a constraint?
Authors need to demonstrate BEq pipeline works across multiple math LLMs to show BEq doesn't only work in specific models, which supports claims about not requiring extensive efforts in specific prompt designs (lines 66-68)
3) In the **Human Equivalence Benchmark**, There are some details requiring clarification to convince readers for fairness of BEq:

   - The mechanism for creating "ground truths": whether through expert annotation, rule-based generation, or other techniques - remains unclear. If experts are involved, the potential overlap between pair creators and equivalence evaluators raises concerns about evaluation independence. The justification for using only 200 cases appears insufficient given Con-NF's larger scale of 900+ cases. Moreover, the validation of ground truth existence in ProofNet needs more rigorous documentation.
   - This part suffers from insufficient documentation of expert qualifications and procedures. The paper fails to specify whether experts were students, authors, or third-party experts, and their required expertise level. Critical operational details are missing, including payment structures, annotation guidelines, and the workflow for equivalence labels. The process for handling cases with multiple valid ground-truths remains ambiguous, as does the distinction between "expert label equivalence" and "ground-truths."
   - The benchmark shows concerning signs of potential sampling bias. The non-uniform sampling evident in Figure 3 requires better explanation, and the paper doesn't adequately address how bias was controlled in LLM-generated examples. The O1 bias control mechanism and prompting strategies need clearer documentation. Additionally, the absence of label distribution information makes it difficult to evaluate the fairness of BEq.

These limitations significantly impact the trust of implementing BEq based on specific LLM for such task.

4) Potential biased benchmarking: I think one of the biggest concern along the entire paper is that it seems the specific LLM InterLM-math conducts the whole evaluation, method backbones. The family of Large Language Models (LLMs) trained on similar datasets inevitably shares inherent biases, which affects evaluation consistency. Did BEq, driven by InterLM-math, gain an unfair advantage in Authoformalization performance if it leverages the same InterLM-math framework? I think a more direct solution to resolve this concern is to show results that BEq implemented by different math LLMs.

### Method:

While Section 4 identifies "nonexistent definitions in library" as a main source of hallucinations and proposes RAG-based solutions, it requires empirical validation of improvement. Table 3 shows overall performance gains but doesn't isolate how much comes from reducing such hallucinations. The paper needs quantitative analysis or visualization demonstrating the specific impact of RAG methods on how effectiveness of hallucination reduction.

### Benchmark:
1) It's nice to see that the authors have annotated a new benchmark, Con-NF, to explore the effectiveness of the OOD capabilities of the developed methods. However, there are no details about benchmark construction or distributions. For example, how was the annotation performed, and what are the topic or problem distributions of the new datasets? Since the authors mention this is an OOD dataset, it would be better to compare the distributions between ProofNet and Con-NF in terms of objectives and disciplines to illustrate how they differ as OOD datasets. I think it’s insufficient to simply mention that one is for under-graduate level and the other for research level, as there is a possibility that they share similar foundational theorems or objects and dependencies or overlapped.

2) Why did the authors choose to create new datasets from scratch instead of utilizing existing ones, such as [1]? It would be more effective for the authors to first specify the features they wish to capture that are not present in existing datasets, thus justifying the need for this curation. If there are existing datasets that already contain the desired OOD features, the reason for annotating a new dataset is less convincing, especially given the lack of detailed descriptions regarding its construction.

### Experiments:
As mentioned, just one backbone math LLM InterLM2-math is not enough. Given a global view of this paper, the metrics are designed specifically by authors and InterLM2-math, the benchmark is annotated by authors, and models in methods are chosen and fine-tuned by the same specific models, the fairness is quite questionable. More datasets and LLM backbones are required in evaluation.

Also more baseline methods and implementations are required. For example:
- Basic reasoning baselines like COT, TOT, Self-Consistency should be compared.
-  [1] implemented ReProver, a RAG method for automated theorem proving in Lean. Differences with workflows in this paper and [1] should be explained and compared. The novelty of this paper appears limited compared to [1] if the performance comparison and analysis are not clarified.
- I don't think it's fair to conduct all fine-tuned methods on one specific model InterLLM2-math, instead, I concern that whether such framework only works for such LLM especially the metric of BEqs is implemented by the same family of LLM (InterLLM also...).

[1] LeanDojo: LeanDojo: Theorem Proving with Retrieval-Augmented Language Models, NeurIPS 2023.

**Questions:**

## Introduction:
1. Why is dependency retrieval essential in the current context, and how does it address existing limitations?
2. What evidence supports the claims about β-reduction and type checking limitations? Can authors provide citations or examples?
3. How does BEq construction overcome the "non-determinant and highly prompt-dependent" limitations of LLM graders when it also uses LLMs and few-shot prompting?

## BEq Evaluator:
1. How does BEq's few-shot prompting actually reduce human design efforts compared to previous approaches?
2. Why choose InternLM2-Math-Plus-20B when: DeepSeek V2.5 shows better performance in majority voting?
3. What makes its math pretraining superior? Why not implement BEq with other models?
4. Can BEq work effectively across different math LLMs?

## Human Equivalence Benchmark:
1. How were ground truth pairs created and validated in details?
2. What are the expert qualifications, payment structures, and annotation guidelines?
3. How was sampling bias controlled in terms of the following aspects:
   - LLM-generated examples?
   - non-uniform distribution in Figure 3?
   - O1 bias control mechanisms to make sure the quality of evaluation?

## Method:
1. How effectively do RAG methods specifically reduce hallucinations in depth?
2. Can authors quantify the improvement in Table 3 attributed to hallucination reduction for "non-existing dependencies" especially?

## Con-NF Questions:
1. What are the construction details and distributions of Con-NF?
2. How do ProofNet and Con-NF differ in terms of discipline, problem distributions and theorem or dependencies?
3. Why create new datasets instead of using existing ones like [1]? What are exact features considered as OOD comparing to exisitng benchmarks so that authors have to spend efforts in annotating a new benchmark?

## Experimental Questions:
1. How would results differ with multiple LLM backbones?
2. Why weren't standard reasoning baselines (COT, TOT, Self-Consistency) included? Comparing with such basic reasoning mehtods is crucial.
3. How does this work differ from ReProver [1]?
4. Does the framework only work effectively with InterLLM2-math? More implementations of different families of LLMs with BEqs are required.

---

> ### Author Response · Authors · 2024-11-22
>
> We greatly appreciate your detailed reviews, as well as your constructive comments. There are too many points in the _Weakness_ and _Question_ sections. For clarity, we have omitted the repeated points in the _Weakness_ section and will address the remaining points in the _Question_ section. If there are any missing questions, please point them out, and we will respond promptly.
>
> ***Weaknesses:***
>
> >***Authors state some disadvantages of "LLM graders". Could authors provide more citations stating this?***
> > 1. The paper critiques LLM graders as "non-determinant and highly dependent on prompts" but doesn't sufficiently distinguish how BEq (which also uses LLMs) overcomes these limitations.
> > 2. The few-shot prompting used in BEq seems to face similar prompt-design challenges that the authors criticize.
> > 3. While Table 1 shows improved performance, more analysis is needed to demonstrate that BEq fundamentally addresses the cited limitations of LLM graders raised in the introduction.
>
> LLM grader[4] is a recently published work. As far as we know, BEq is the first work trying to address the non-deterministic, uninterpretability, and prompt-dependency of LLM Grader. However, an LLM being "non-deterministic and highly dependent on prompts" seems to be common knowledge in the LLM and DL community, and many methods are even built on this "non-deterministic" nature, for example, Majority Voting.
>
> We've invested significant effort in reproducing the LLM grader and establishing it as a strong baseline. The 82.5% accuracy reported in Table 1 represents the best result among our experiments. Below are some findings from our preliminary experiments on the base model and prompt method used in the LLM Grader. We've also explored other base models and methods, recordings of the best results obtained.
>
> | Model                   | Method            | Precision | Recall | Accuray |
> | ----------------------- | ----------------- | --------- | ------ | ------- |
> | Qwen1.5-14B-Chat        | Original (1-shot) | 42.64%    | 78.57% | 55.50%  |
> | internlm2-math-plus-20b | MajVot (16-shot)  | 37.28%    | 91.30% | 44.00%  |
> | Deepseek-V2.5           | MajVot (16-shot)  | 70.59%    | 85.71% | 82.50%  |
>
> It seems that you might have misunderstood BEq as a prompt-based method. In fact, BEq defines equivalence as _bidirectionally "convertible" using simple transformations_. Few-shot prompting LLM is just an implementation to approximate the ideal transformation function $T$. To assert that "$\boldsymbol s_P$ is BEq to $\boldsymbol s_Q$", one should provide (or the LLM is prompted to generate) two proofs (tactic sequence) of UDI (unidirectional definitional implication):
>
> 1. Proof of $\boldsymbol s_Q$ assuming $\boldsymbol s_P$ to be `True`, and this proof is limited to only use tactics in $\mathcal R$ and must use the hypothesis that $\boldsymbol s_Q$ holds.
> 2. Proof of $\boldsymbol s_P$ assuming $\boldsymbol s_Q$ to be `True`, and this proof is limited to only use tactics in $\mathcal R$ and must use the hypothesis that $\boldsymbol s_P$ holds.
>
> These proofs are interpretable and formal-verified. This approximation becomes more effective as the base LLM becomes more capable or the number of attempts $K$ increases (discussed in Appendix A.1).
>
> However, despite its interpetability, accuracy and formal-verified nature, we couldn't overclaim that BEq fundamentally addresses all limitations of existing methods. As described in _Sec.6.Limitations of BEq_, carefully designing the limitation of transformation primitives $\mathcal R$ and improving the LLM's capabilities to approximate $T$ is crucial. We sincerely invite community efforts to delve into refining BEq and eventually set a domain standard to facilitate subsequent research.

---

> ### Author Response · Authors · 2024-11-22
>
> ***Questions:***
>
> **Introduction:**
>
> >***Why is dependency retrieval essential in the current context, and how does it address existing limitations?***
>
> As stated in line 070-075 and 090-100 of the paper, the current autoformalization paradigm suffers from the agnosia of context and dependency retrieval helps address this issue.
>
> Due to the lack of prior knowledge of existing formalized definitions and theorems, current autoformalization models frequently generate hallucinated formal objects that do not exist in the library.
>
> Dependency retrieval addresses this issue by retrieving potentially dependent formal objects from the formal library (e.g., Mathlib 4) conditioned on the informal statement to autoformalize. Given these retrieved objects, the autoformalizer model can generate more accurate formal statements with lower risk of hallucination.
>
>
> >***What evidence supports the claims about β-reduction and type checking limitations? Can authors provide citations or examples?***
>
> Actually, we did not mention anything about the limitations of $\beta$-reduction in the paper. It seems that you need examples about the fragility of machine translation metrics (like BLEU) under $\beta$-reduction. Suppose we define a function `def dummy (x : Nat) := 1`. Then, the two expressions, `dummy x` and `1` are definitionally equal, but they have low token-level similarity, and thus struggle to be viewed "equivalent" in machine translation metrics like BLEU.
>
>
> As for the **limitation of existing metrics such as typecheck, BLEU and LLM Grader**, please kindly refer to the two demonstrations in Fig.8 and Fig.9 of the paper, where all of the metrics fail. Here we provide a brief analysis of Fig.8. Its informal statement is "Show that $\sin\frac{\pi}{12}$ is an algebraic number", and two formal statments be `isAlgebraic Rat (sin (pi/12))` and `isAlgebraic Rat (Real.sin (Real.pi/12))`. These two formal statements can both pass the Lean typecheck, have high token-level similarity and are very confusing for LLM (and people who is not proficient in Lean 4). However, they're completely different in Lean. `pi` in the former statement is an implicit argument of an arbitrary real number, while `Real.pi` in the latter statement means the true $\pi$.
>
> **BEq Evaluator:**
>
> >***How does BEq construction overcome the "non-determinant and highly prompt-dependent" limitations of LLM graders when it also uses LLMs and few-shot prompting?***
> >***How does BEq's few-shot prompting actually reduce human design efforts compared to previous approaches?***
>
> Answers to these two questions are included in the response in the _Weakness_ section. Please kindly refer to it.
>
> >***Why choose InternLM2-Math-Plus-20B when: DeepSeek V2.5 shows better performance in majority voting? What makes its math pretraining superior? Why not implement BEq with other models? Can BEq work effectively across different math LLMs?***
>
> Interesting question. There is no special reason for us to choose InternLM2-Math-Plus-20B. Maybe the most important one is that the API calling of Deepseek V2.5 costs money.
> To better answer this question, we have evaluated BEq on different models with attempt number $k=8$ and temperature $T=0$ (For Deepseek-V2.5, $T=0.7$ due to the requirement of API calling), and the results are as follows.
>
> | Model                           | Size | Precision | Recall | Accuray |
> | ------------------------------- | ---- | --------- | ------ | ------- |
> | internlm2-math-plus-7b          | 7B   | 100.00%   | 72.86% | 90.50%  |
> | Qwen1.5-14B-Chat                | 14B  | 100.00%   | 67.14% | 88.50%  |
> | DeepSeek-Coder-V2-Lite-Instruct | 16B  | 100.00%   | 71.43% | 90.00%  |
> | internlm2-math-plus-20b         | 20B  | 100.00%   | 72.86% | 90.50%  |
> | Deepseek-V2.5                   | 236B | 100.00%   | 68.57% | 89.00%  |
>
> As the results indicate, **a 7B InternLM2-math-plus model can achieve 90.5% accuracy**. There is not much difference among the various models, but the pretraining dataset and model architecture do play a significant role. Qwen1.5, which likely was not pretrained on a vast formal math corpus, shows inferior accuracy compared to all other models. In contrast, Deepseek-series and InternLM-series models exhibit slightly higher accuracy. Additionally, **we believe there is substantial room for improving the LLM, potentially through data synthesis and fine-tuning, as the recall is not yet satisfactory.**

---

> ### Author Response · Authors · 2024-11-22
>
> **Human Equivalence Benchmark:**
>
> >***How were ground truth pairs created and validated in details?***
>
> As detailedly described in Sec.3.2 and Appendix A.4, we randomly sample 200 successfully typechecked model predictions on the ProofNet[1] benchmark: 1) 100 from an early version of the RAutoformalizer; 2) 100 from closed-source ICL prompted OpenAI o1-preview.
>
> These predictions are generated with hyperparameters:
> 1) greedy decoding with $T=0.0$ for RAutoformalizer;
> 2) temperature decoding with default hyperparameters of OpenAI API for o1-preview.
>
> Then, 4 experts, one from formal verification and three from computer science, are invited to label the model generated formal statement as “equivalent” or “inequivalent” with the corresponding ground-truth statements.
>
> Experts first separately evaluate the equivalence, and finally discuss in round-table to reach an agreement for each sample.
>
> >***What are the expert qualifications, payment structures, and annotation guidelines?***
>
> Thank you for your careful review. More details about the construction of human equivalence benchmark have been added in our revised version in **Appendix A.4 Human Equivalence Benchmark**. The expert group consists of 1) one professor in formal verification; 2) one PhD student in EECS; 3) two master students in CS. However, In accordance with the principles of anonymity, we regret to inform you that we cannot disclose any additional information. As stated in Sec.7 Reproducibility Statement, the human equivalence benchmark will be released after acceptance and whoever interested can check then.
>
> >***How was sampling bias controlled in terms of the following aspects:***
> > - LLM-generated examples?
> > - non-uniform distribution in Figure 3?
> > - O1 bias control mechanisms to make sure the quality of evaluation?
>
> Good question. Actually, we've tried our best to control the biases of the Human Equivalence Benchmark and make it suitable for general cases.
>
> 1. We collect model predictions from both few-shot prompting closed-source OpenAI o1-preview and fine-tuning InternLM-Math-Base-7B (an early version of RAutoformalizer) to avoid biases introduced in 1) Generation mode (ICL vs. Fine-tuning); 2) Model size; 3) Open-source model and closed-source model.
>
> 2. The non-uniform distribution in Fig. 3 can be attributed to two aspects:
>     1. The original discipline distribution of ProofNet is imbalanced. Its distribution is as follows:
>         - Analysis: 88
>         - Examinations: 15
>         - Linear Algebra: 28
>         - Number Theory: 21
>         - Topology: 60
>         - Abstract Algebra: 162
>     2. As detailed in line 296, 200 formal statements are uniformly sampled from the typechecked predictions generated by RAutoformalizer and OpenAI's o1-preview (100 predictions from each). Since the ratio of samples to pass typecheck differs between disciplines, the distribution becomes more unbalanced.
>
> 3. It seems that we didn’t mention anything about the "O1 bias control mechanisms". Actually, the prompts we used for OpenAI o1-preview, GPT-4o, and Deepseek-V2.5 to perform ICL autoformalization are shown in Appendix A.7.3.
>
> If you still feel that the Human Equivalence Benchmark may not fully address your concerns, please refer to Appendix A.2.1, where the results of our manual evaluation may provide further insights into BEq and RAutoformalizer.
>
> **Method:**
>
> >***How effectively do RAG methods specifically reduce hallucinations in depth? Can authors quantify the improvement in Table 3 attributed to hallucination reduction for "non-existing dependencies" especially?***
>
> Sure, we quantitatively analyzed it by delving into the error distribution of autoformalizations that do not pass typecheck. We parsed the Lean error messages of autoformalizations that fails to typecheck and coarsely classified them into hallucination errors and other errors (detailed taxonomy can be found in Appendix A.2.3). The statistics are reported below:
>
> |               | **ProofNet**      | -              | **Con-NF**        | -              |
> | ------------- | ----------------- | -------------- | ----------------- | -------------- |
> | **Method**    | **Hallucination** | **Others**     | **Hallucination** | **Others**     |
> | **RA -R**     | 434               | 1790           | 8902              | 14842          |
> | **RA**        | 320 (-26.27%)     | 1500 (-16.20%) | 5217 (-41.40%)    | 13386 (-9.81%) |
> | **RA +R**     | 65 (-85.02%)      | 1173 (-34.47%) | 1134 (-87.26%)    | 5882 (-60.37%) |
>
> Results show that both types of errors are reduced by Dependency Retrieval, especially Hallucination errors. This might be one of the root causes of RAutoformalizer’s ablative improvement.

---

> ### Author Response · Authors · 2024-11-22
>
> **Con-NF Questions:**
>
> >***What are the construction details and distributions of Con-NF?***
>
> The construction process is described in detail in _Sec.4.1 Con-NF: OOD Benchmark_. During construction, rejective sampling and post-informalization filtering policies are applied in benchmark construction, including:
> 1. Length filtering: Generated informal statements which are too long or too short are rejected.
> 2. Format filtering: Generated informal statements which do not match the given format are rejected.
> 3. Special token filtering: Generated informal statements which contain some special tokens (which we think to be a signal of low-quality informalization) are rejected.
> 4. Semantic filtering: Sophisticated rules are applied to filter out samples with large semantic gaps with the corresponding math object.
> 5. Source filtering: Samples whose all dependencies are in Mathlib 4 are filtered out to ensure the OOD nature of this benchmark.
>
> Moreover, human supervision is conducted on statements that fail to pass the filters after several samplings. The sophisticated filtering process reduces the size of the Con-NF benchmark from 85762 to 961.
>
> The discipline distribution of statements in Con-NF benchmark are 100% on Quine's set theory _New Foundations_, which is not included in Mathlib 4.
>
> >***How do ProofNet and Con-NF differ in terms of discipline, problem distributions and theorem or dependencies?***
>
> ProofNet is a benchmark for autoformalization on **undergraduate-level mathematics**, with the following problem source distribution:
> - Analysis: 88
> - Examinations: 15
> - Linear Algebra: 28
> - Number Theory: 21
> - Topology: 60
> - Abstract Algebra: 162
>
> **All formal statements in ProofNet totally depends on Mathlib 4** (only about ~10 of them slightly dependes on user-defined function or abbreviation). Mathlib 4 is a user maintained library with both programming infrastructure and mathematics, as well as tactics that use the former and allow to develop the latter. Many Lean users view Mathlib as a "standard library" of Lean.
>
> However, the Con-NF benchmark is constructed on [Con(NF)](https://github.com/leanprover-community/con-nf), a formal consistency proof of Quine's set theory _New Foundations_. After our filtering, statements in this benchmark **all belong to this theory** and have **novel dependencies that don't exist in Mathlib 4**.
>
>
> >***Why create new datasets instead of using existing ones like ReProver?***
>
> Because this paper focuses on a **completely different task than ReProver**. This paper concentrates on _statement autoformalization_, translating a math statement from natural language to formal language, while ReProver focuses on _theorem proving_, searching for a formal-verified proof given a formal statement.
>
> > ***What are exact features considered as OOD comparing to exisitng benchmarks so that authors have to spend efforts in annotating a new benchmark?***
>
> RAutoformalizer is trained on statements in Mathlib 4, for which all dependencies are also included in Mathlib 4. However, for the Con-NF benchmark, all statements are **based on a novel theory** that is not included in Mathlib 4 and **have novel dependencies that don't exist in Mathlib 4**.

---

> ### Author Response · Authors · 2024-11-22
>
> **Experimental Questions:**
>
> >***How would results differ with multiple LLM backbones?***
>
> We have reproduced all fine-tuning-based methods on DeepseekMath-Base-7B as the base model with training recipes identical to Appendix A.8. The results are added to Appendix A.3, which demonstrate consistent (and even clearer) advantage of our methods over all baselines, and the ablative improvement of Dependency Retrieval.
>
>
> >***Why weren't standard reasoning baselines (COT, TOT, Self-Consistency) included? Comparing with such basic reasoning mehtods is crucial.***
>
> Because our task, statement autoformalization, which translates a math statement from natural language to formal language, resembles more to machine translation than reasoning. None of the previous autoformalization literatures (including but not limited to [1-4,6]) have included prompt-engineering methods such as CoT and ToT in their baselines. A recent work [5] (later than our submission) proposes an implementation of self-consistency on the Isabelle theorem prover instead of Lean. However, reproducing it on Lean faces many technical challenges and mismatches of language features.
>
> We have hand-crafted a CoT implementation of autoformalization, and the evaluation results are as follows:
>
> | Benchmark     |         | ProofNet    |       |             |        | Con-NF      |       |             |       |
> | ------------- | ------- | ----------- | ----- | ----------- | ------ | ----------- | ----- | ----------- | ----- |
> | Base Model    | Method  | Typecheck@1 | BEq@1 | Typecheck@8 | BEq@8  | Typecheck@1 | BEq@1 | Typecheck@8 | BEq@8 |
> | Deepseek V2.5 | ICL     | **40.37%**      | **9.89%** | 51.07%      | 10.96% | 9.37%       | **2.81%** | 16.23%      | **4.27%** |
> |              | ICL+CoT | 33.16%      | 6.68% | **62.57%**      | **16.84%** | **10.72%**      | 1.46% | **24.04%**      | 3.85% |
>
> Results show that the results of CoT is unstable. CoT might be better on in-distribution, multiple-trial settings.
>
> >***How does this work differ from ReProver?***
>
> 1. RAutoformalizer addresses a **completely different task to ReProver**. RAutoformalizer concentrates on _statement autoformalization_, translating a math statement from natural language to formal language, while ReProver focuses on _theorem proving_, searching for a formal-verified proof given a formal statement.
> 2. The **retrieval tasks are completely different**. We firstly propose to augment autoformalization by _dependency retrieval_, retrieving **dependent math objects** from a formal library **given a natural language statement**, while ReProver augments theorem provers by _premise selection_, retrieving **applicable premises** from a formal library **given a formal proof state**.
>
> >***Does the framework only work effectively with InterLLM2-math? More implementations of different families of LLMs with BEqs are required.***
>
> RAutoformalizer is not limited to InternLM2-Math. We have reproduced all fine-tuning-based methods on DeepseekMath-Base-7B as the base model with training recipes identical to Appendix A.8. The results are as follows, which demonstrate consistent (and even clearer) advantage of our methods over all baselines, and the ablative improvement of Dependency Retrieval.
>
>
> | **Benchmark** | **ProofNet**    | -          | -          | -          | **Con-NF** | -         | -          | -          |
> | ------------- |:--------------- | ---------- | ---------- | ---------- | ---------- | --------- | ---------- | ---------- |
> | **Method**    | **Typecheck@1** | **BEq@1**  | **Typecheck@8**    | **BEq@8**  | **Typecheck@1**    | **BEq@1** | **Typecheck@8**    | **BEq@8**  |
> | MMA           | 15.78%          | 1.87%      | 31.02%     | 5.08%      | 3.23%      | 1.66%     | 7.28%      | 4.06%      |
> | MMA (L)       | 17.65%          | 2.41%      | 31.02%     | 5.61%      | 2.71%      | 1.35%     | 7.39%      | 4.37%      |
> | PDA           | 14.71%          | 2.14%      | 27.54%     | 5.61%      | 4.89%      | 1.77%     | 10.82%     | 4.47%      |
> | LW            | 36.10%          | 8.56%      | 53.74%     | 10.16%     | 4.89%      | 1.98%     | 11.13%     | 2.08%      |
> | **RA -R**     | 51.34%          | 10.96%     | 69.79%     | 15.24%     | 8.22%      | 3.12%     | 12.59%     | 4.27%      |
> | **RA**        | **59.36%**      | **10.96%** | **75.94%** | **17.91%** | **17.59%** | **9.68%** | **25.49%** | **15.30%** |
> | **RA +R**     | 72.73%          | 23.80%     | 83.69%     | 32.62%     | 60.56%     | 44.02%    | 75.96%     | 59.00%     |
>
>
> Thanks for your suggestion. We've added this experiment to Appendix A.3.

---

> ### Author Response · Authors · 2024-11-22
>
> [1] Azerbayev, Zhangir, et al. "Proofnet: Autoformalizing and formally proving undergraduate-level mathematics." _arXiv preprint arXiv:2302.12433_ (2023).
>
> [2] Wu, Yuhuai, et al. "Autoformalization with large language models." Advances in Neural Information Processing Systems 35 (2022): 32353-32368.
>
> [3] Jiang, Albert Q., Wenda Li, and Mateja Jamnik. "Multilingual mathematical autoformalization." arXiv preprint arXiv:2311.03755 (2023).
>
> [4] Ying, Huaiyuan, et al. "Lean Workbook: A large-scale Lean problem set formalized from natural language math problems." arXiv preprint arXiv:2406.03847 (2024).
>
> [5] Li, Zenan, et al. "Autoformalize Mathematical Statements by Symbolic Equivalence and Semantic Consistency." arXiv preprint arXiv:2410.20936 (2024).
>
> [6] Lu, Jianqiao, et al. "Process-driven autoformalization in lean 4." arXiv preprint arXiv:2406.01940 (2024).

---

> > ### Comment · Reviewer_6DKY · 2024-11-25
> > **Thanks for answering my question**
> >
> > After carefully reading, I still keep concerns about bias raised in few-shot prompt designing for evaluators and especially since the whole workflow are dominated by LLMs. Also, authors are making contributions to Math-based work, but require readers to have a background of neural machine translation, which hurts readability pretty much. However, I'm glad to see more detailed and rigorous interpretations about their benchmark set-ups and more experiments to further validate their evaluations. Even though I keep doubt about its fairness, it did make the step further to autoformulations. I'm satisfied with the revised version, which becomes more rigorous and makes more sense. I have adjusted my score.

---

> > > ### Author Response · Authors · 2024-11-26
> > >
> > > Many thanks for your time and efforts in reviewing, and the appreciation of our work.
> > >
> > > One of the future works of BEq is establishing a more robust and widely-recognized evaluation method for statement autoformalization, for which we sincerely invite collaboration and suggestions from the whole community :)

---

### Official Review · Reviewer_9PuK · 2024-11-02

**Soundness:** 3
**Presentation:** 4
**Contribution:** 3
**Rating:** 8
**Confidence:** 4

**Summary:**

The authors present three contributions to the field of autoformalization. First, a new evaluation metric (BEq) which better aligns with human evaluation compared to classic Machine Translation metrics. This is done by extending the concept of definitional equivalence, which can be overly strict in certain cases and not allow enough flexibility to accept all possible correct answers. Second, the authors introduce a retrieval-augmented autoformalization method which retrieves necessary dependencies for a theorem before formalization. Finally, the authors introduce a new task (dependency retrieval) and associated benchmark (Con-NF) to help measure performance for research-level autoformalization.

**Strengths:**

The authors work on an important problem in autoformalization, namely the lack of flexible automatic metrics. Their approach is well-explained and well-motivated, and I believe it can be of significant use to the community. BEq significantly improves over other metrics in terms of human-alignment. I believe this is the strongest contribution of the paper.

Similarly, the dependency retrieval task is a fundamental task in autoformalization, since otherwise we expect LLMs to be fluent in all necessary context, which is impossible for research-level mathematics which are introduced after the LLM was trained. Thus, the introduction of the task together with a benchmark and baseline approaches is a strong contribution.

Overall I think this paper is a solid contribution to the community and is well-supported by the experiments.

**Weaknesses:**

While the BEq metric improves on previous metrics, one of its shortcomings is that it requires gold label formal statements (compared to, for example, type checking, which only requires the input data). Both training data and benchmarks are lacking in this area, making BEq less impactful overall.

Additionally, BEq requires a 20b parameter model to execute, severely restricting access to the metric for researchers without the necessary compute. A significant improvement would be reducing this model size, even to a 7b size which would make inference possible on much smaller devices and with improved efficiency. I didn't notice any performance comparison between various model sizes, which could improve the stance of this model choice.

**Questions:**

Have you tested different model sizes for the BEq metric? If so, what performance drop-off did you see with smaller models? An analysis of different failure cases for these models would also be very insightful, and could offer avenues for improvement.

---

> ### Author Response · Authors · 2024-11-22
>
> Thank you for the appreciation of our work and the inspiring suggestions. We sincerely hope the following explanations can help resolve your concerns.
>
> > Weekness1. BEq requires gold label formal statements.
>
> Admittedly, this is one of the biggest limitation of BEq, as well as an **open problem of the whole ML community** for years. It is difficult to evaluate a task of the form $\boldsymbol y=f(\boldsymbol x)$ without a ground-truth $\boldsymbol y$. All discriminative tasks need to compare predictions with gold ground-truth labels. For generative tasks, to coarsely summarize, there are mainly 4 types of automated evaluation methods:
> 1. Using learning-based metrics, such as FID (Fréchet inception distance) in image generation tasks;
> 2. Using symbolic metrics, such as calculator or formal verifier (in theorem proving);
> 3. Partial evaluation. For example, existing informal reasoning benchmarks such as MATH[2] and MMLU[1] only perform identity matching between model-predicted answers and the ground-truth answers, neglecting the reasoning trajectory;
> 4. Proxy metrics, such as BLEU in machine translation.
>
> 1,3,4 either depend on ground-truth labels or an “evaluator” DNN, which is uninterpretable and vulnerable to adversarial attacks. And 2 is not suitable for our task, because the space of free-form natural language statements in autoformalization is intractable for symbolic verifier.
> We currently have no idea of this limitation and believe solving this can lead to strong contribution for the whole ML community.
> As for autoformalization, **all existing benchmarks (e.g. ProofNet[3], FormL4[4]) have ground-truth formal statements**. And it should be the benchmark authors' responsibility to ensure the quality of ground-truth.
>
> > Weekness2. BEq requires a 20b parameter model to execute.
>
> To address your concern, we have evaluated BEq on different models with beam search using temperature $T=0$, generation number $k=8$ (samples 8 tactic sequences candidates for each input). For Deepseek-V2.5, we use temperature sampling with $T=0.7$ due to the limitation of API call. The results are as follows.
>
> | Model                           | Size | Precision | Recall | Accuray |
> | ------------------------------- | ---- | --------- | ------ | ------- |
> | internlm2-math-plus-7b          | 7B   | 100.00%   | 72.86% | 90.50%  |
> | Qwen1.5-14B-Chat                | 14B  | 100.00%   | 67.14% | 88.50%  |
> | DeepSeek-Coder-V2-Lite-Instruct | 16B  | 100.00%   | 71.43% | 90.00%  |
> | internlm2-math-plus-20b         | 20B  | 100.00%   | 72.86% | 90.50%  |
> | Deepseek-V2.5                   | 236B | 100.00%   | 68.57% | 89.00%  |
> As the results indicate, **a 7B InternLM2-Math-Plus model can achieve 90.5% accuracy**. There is not much difference between the various models, but the pretraining dataset and model architecture do play a significant role. Qwen1.5, which likely was not pretrained on a vast formal math corpus, shows inferior accuracy compared to all other models. In contrast, Deepseek-series and InternLM-series models exhibit slightly higher accuracy. Additionally, we **believe there is substantial room for improving the LLM, potentially through data synthesis and fine-tuning, as the recall is not yet satisfactory.**
>
> > Question1. Experiments of different model sizes for BEq?
>
> Thank you for your constructive comment. The experiment results are detailed in the **response to Weakness2**. During case analysis, We observe that all models share similar failure patterns, as analyzed in **Appendix A.4, Failure Case Analysis**. We find that ICL prompting is relatively weak and fails to generate proper transformations for complex scenarios. Therefore, fine-tuning an expert model might be a viable solution. This work focuses on proposing the neural-symbolic method of BEq and demonstrating the potential of RAutoformalizer. To establish a widely-recognized evaluation standard for statement autoformalization, we sincerely invite community efforts to refine BEq.
>
> [1] Hendrycks, Dan, et al. "Measuring massive multitask language understanding." _arXiv preprint arXiv:2009.03300_ (2020).
>
> [2] Hendrycks, Dan, et al. "Measuring mathematical problem solving with the math dataset." _arXiv preprint arXiv:2103.03874_ (2021).
>
> [3] Azerbayev, Zhangir, et al. "Proofnet: Autoformalizing and formally proving undergraduate-level mathematics." _arXiv preprint arXiv:2302.12433_ (2023).
>
> [4] Lu, Jianqiao, et al. "Process-driven autoformalization in lean 4." arXiv preprint arXiv:2406.01940 (2024).

---

> > ### Comment · Reviewer_9PuK · 2024-11-25
> > **Response to Rebuttal**
> >
> > Thank you authors for the detailed response.

---

> ### Comment · Area_Chair_hUz9 · 2024-11-25
> **Reminder: Rebuttal Deadline for ICLR 2025**
>
> Dear Reviewer 9PuK,
>
> As the rebuttal deadline approaches, please kindly check the papers' discussion threads and respond to the authors' rebuttals. If you haven't had a chance to respond yet, I’d greatly appreciate your input soon. Your insights are invaluable to the authors and the review process.
>
> Thank you for your effort and support!
>
> Best regards,
>
> Area chair

---

### Official Review · Reviewer_YeoP · 2024-11-04

**Soundness:** 3
**Presentation:** 3
**Contribution:** 3
**Rating:** 6
**Confidence:** 2

**Summary:**

This paper aims to improve existing statement autoformalization paradigms by two key limitations: 1. absence of faithful and universal automated evaluation for autoformalization results; 2 agnosia of contextural information. To this end, the paper trains a retriever to retrieve dependent information and a autoformalizer to attempt the proof.

**Strengths:**

1.	The design of the training protocol seems reasonable, like the formal data in existing library is parsed into informal sentences and stored in topological order. The retrieval first and generation next works well for autoformalization.
2.	The experiment result shows that the newly proposed BEq method surpasses baselines by a landslide.
3.	A novel benchmark, Con-NF is built from frontier mathematical researches.

**Weaknesses:**

Some certain details can be included to make the paper more self-contained. Like the reason why the dense embedding is sufficient for the dependency retrieval.

**Questions:**

See weakness.

---

> ### Author Response · Authors · 2024-11-22
>
> Thank you for your helpful suggestions. We have added more details in the revision and hopefully the explanation can resolve your concerns.
>
> > Weekness1. Missing of details.
>
> Thanks for your suggestion, we have added the following details:
> 1. The prompt template details of BEq in Appendix A.7.1;
> 2. The prompt template details of LLM Grader to Appendix A.7.2;
> 3. The prompt template details of ICL autoformalization to Appendix A.7.2;
> 4. More details in constructing the Human Equivalence Benchmark in Appendix A.4;
> 5. More ablative experiments on RAutoformalizer in Appendix A.2.2 and A.2.3;
> 6. A more thorough evaluation of BEq in Appendix A.2.1;
> 7. Reproduction experiments of all fine-tuning-based methods on DeepseekMath-7B in Appendix A.3;
> 8. Ablative experiments of augmenting ICL methods in Appendix A.10.
>
> Hopefully, these details can enhance reliability, logical coherence and reproducibility of this work.
>
> > Weekness1. Explanation of why the dense embedding is sufficient.
>
> We adopt dense embedding for the following reasons:
>
> 1. Dense embedding is one of the simplest and most prevalent method used in RAG [1,2].
> 2. ReProver[3] adopts dense embedding in premise selection tasks, and we follow their choice in dependency retrieval.
> 3. We do not conclude that dense embedding is sufficient in the paper. On the contrary, we believe there is significant room for improvement, as indicated by the metrics in Table 2 and the comparison between RA and RA (+R) in Table 3. In this paper, our focus is on proposing the task of dependency retrieval and demonstrating its importance. We leave more sophisticated upgrades, such as multi-vector embeddings, re-ranking, and query augmentation, to the community, as mentioned in Section 6 *Limitations of RAutoformalizer*.
>
> [1] Zhao S, Yang Y, Wang Z, et al. Retrieval Augmented Generation (RAG) and Beyond: A Comprehensive Survey on How to Make your LLMs use External Data More Wisely[J]. arXiv preprint arXiv:2409.14924, 2024.
>
> [2] Gao Y, Xiong Y, Gao X, et al. Retrieval-augmented generation for large language models: A survey[J]. arXiv preprint arXiv:2312.10997, 2023.
>
> [3] Yang, Kaiyu, et al. "Leandojo: Theorem proving with retrieval-augmented language models." _Advances in Neural Information Processing Systems_ 36 (2024).

---

> > ### Comment · Reviewer_YeoP · 2024-11-26
> >
> > Dear authors, thank you for your rebuttal and I will keep my score as it's already suggest for accept now.

---

> ### Comment · Area_Chair_hUz9 · 2024-11-25
> **Reminder: Rebuttal Deadline for ICLR 2025**
>
> Dear Reviewer YeoP,
>
> As the rebuttal deadline approaches, please kindly check the papers' discussion threads and respond to the authors' rebuttals. If you haven't had a chance to respond yet, I’d greatly appreciate your input soon. Your insights are invaluable to the authors and the review process.
>
> Thank you for your effort and support!
>
> Best regards,
>
> Area chair

---

### Official Review · Reviewer_fV7x · 2024-11-04

**Soundness:** 4
**Presentation:** 4
**Contribution:** 3
**Rating:** 8
**Confidence:** 3

**Summary:**

The paper addresses a few key challenges in the problem of LLM-based AutoFormalization. Firstly, they introduce BEq - a new metric to determine equivalence of formal statements -allowing for a better quantification of success for the task. Secondly, they propose RAutoformalizer a new autoformalization technique that retrieves dependent formal objects using informal statements. Lastly, to quantify out of distribution generalization, they propose a new benchmark for the task - Con-NF. They empirically demonstrate that BEq provides expert level performance on determining semantic equivalence of formal statements and then report significantly better results using RAutoformalizer compared to existing SoTA approaches.

**Strengths:**

1. The paper is very well-written, easy to follow and well-organized.
2. The paper presents various key contributions to the Autoformalizer literature in the form of BEq and RAutoformalizer - which can have a great impact research in this area significantly.
3. The BEq metric solves a key challenge in determining equivalence and is a promising alternative to contemporary techniques. I forsee more future work in the literature using this as an importatnt metric of success.
4. The RAutoformalizer technique outperforms current SoTA Autoformalization techniques significantly.

**Weaknesses:**

While I don't see any major weakness in the paper,there are a few minor concerns that maybe worth looking at:
1. BEq as a metric is quite promising, it is not very clear how does it fundamentally align with human judgements. It may be interesting to go a bit deeper into this in the paper.
2. It is not very clear what kinds of benefits the retrieval component in RAutoformalizer is bringing and what are existing gaps from the ground truth retrieval setup.
3. The ICL baselines are a bit too week. It would have been interesting to see how frontier models like DeepSeek and GPT-4o perform with the retrievals provided to them in-context as well.

**Questions:**

Please refer to the Weakness section.

---

> ### Author Response · Authors · 2024-11-22
>
> Thank you for your appreciation of this work.
>
> > Weakness1. More discussions on BEq are required.
>
> Intuitively, equivalence in human perspective is generally one that can be quickly determined and reasoned. Therefore, the key lies in how to define an equivalence relation that can be demonstrated through brief proofs. Thus, we choose to build BEq by extending definitional equality with limited tactical proofs
>
> Then we analyze BEq's limitation.
>
> BEq is based on three central components, **definitional equality**, **proving** and **tactic restriction**. We separately discuss them and list corresponding potentially controversial points as follows.
> - *Definitional Equality* in Lean is an equivalence relation under a series of conversion rules[1][2]. Some of them are quite intuitive, such as *$\beta$-reduction* (function application), *$\zeta$-reduction* (eliminating let-in definitions), *$\delta$-reduction* (unfolding variable and constant definitions). But some might be controversial:
> 	1. *$\alpha$-conversion*, which means two formal statements are equivalent to renaming bound variables. For example, `∀ x : Nat, x + 1 > x` is definitionally equivalent to `∀ y : Nat, y + 1 > y`.
> 	2. Nat literal reduction, which reduces all natural numbers as well as related operations to the final result. For example, `1 + 2 + 3` reduces to `6`, and `6` also reduces to `6`, thus statements `1 + 2 + 3 = 6` and `6 = 6` are definitionally equal.
> - Proving.
> 	- Introducing satisfiable dummy variables. For example, `(x : Nat) : x + 1 > x` and `(x y : Nat) : x + 1 > x` are equivalent under BEq since they can prove each other by adding a dummy parameter or omitting a parameter. Significantly, if the introduced variables are unsatisfiable (of some empty type), the modified statements are not equivalent anymore, e.g. `(x : Nat) : x + 1 > x` is inequivalent to `(x : Nat) (hN : False) : x + 1 > x`.
> 	- Some trivial logically equivalent statements do not differentiate in BEq. For example, `a * b = b ↔ a = 1` and `a * b ≠ b ↔ a ≠ 1`.
> - *Tactic restriction*.
> 	Tactics must be carefully restricted, with the related experiments detailed in Appendix A.1.
>
> Several failure cases and success cases of BEq can be found in Appendix A.4.
> We also more comprehensively compare BEq with expert labels in multiple settings in Appendix A.2.1. The results highlight BEq's robustness on Con-NF, reaching nearly 100% accuracies.
>
> > Weakness2. More ablative analysis of RAutoformalizer is required.
>
> Intriguing question. We quantitatively analyzed the ablative improvement of dependency retrieval by delving into the error distribution of autoformalizations that do not pass typecheck. We parsed the Lean error messages in the un-typechecked autoformalizations and coarsely classified them into hallucination errors and other errors (detailed taxonomy can be found in Appendix A.2.3). The statistics are reported below:
>
> |               | **ProofNet**      | -              | **Con-NF**        | -              |
> | ------------- | ----------------- | -------------- | ----------------- | -------------- |
> | **Method**    | **Hallucination** | **Others**     | **Hallucination** | **Others**     |
> | **RA -R**     | 434               | 1790           | 8902              | 14842          |
> | **RA**        | 320 (-26.27%)     | 1500 (-16.20%) | 5217 (-41.40%)    | 13386 (-9.81%) |
> | **RA +R**     | 65 (-85.02%)      | 1173 (-34.47%) | 1134 (-87.26%)    | 5882 (-60.37%) |
>
> Results show that both types of errors are reduced by Dependency Retrieval, especially Hallucination errors. This might be one of the root causes of RAutoformalizer's ablative improvement.
>
> This discussion is added to _Appendix A.2.3_.

---

> ### Author Response · Authors · 2024-11-22
>
> > Weakness3. Experiments on Dependency-retrieval-augmented ICL LLMs are required.
>
> Sure. Thank you for the constructive suggestion. We have evaluated the performance of augmenting ICL methods by Dependency Retrieval, and the results are as follows. `ICL` denotes the vanilla in-context learning method, `ICL+RA` denotes in-context learning with retrieval-augment (retrieve $5$ dependencies for each statement using fine-tuned BGE-M3 described in Sec.4.2) and `ICL+RA(+R)` denotes in-context learning augmented by ground-truth dependencies.
>
> | **Benchmark**     | -           | **ProofNet** | -         | -          | -          | **Con-NF** | -         | -          | -          |
> | ----------------- | ----------- | ------------ | --------- | ---------- | ---------- | ---------- | --------- | ---------- | ---------- |
> | **Base Model**    | **Method**  | **T@1**      | **Beq@1** | **T@8**    | **Beq@8**  | **T@1**    | **Beq@1** | **T@8**    | **Beq@8**  |
> | **GPT-4o**        | ICL         | 43.58%       | **7.22%** | 66.31%     | 12.83%     | 9.78%      | 1.46%     | 20.71%     | 4.16%      |
> | -                 | ICL+RA      | **46.52%**   | 6.95%     | **77.01%** | **13.37%** | **22.79%** | **6.66%** | **50.57%** | **12.59%** |
> | -                 | ICL+RA (+R) | 58.56%       | 17.38%    | 81.28%     | 29.14%     | 54.84%     | 38.40%    | 75.75%     | 54.11%     |
> | **Deepseek V2.5** | ICL         | 40.37%       | **9.89%** | 51.07%     | **10.96%** | **9.37%**  | **2.81%** | **16.23%** | **4.27%**  |
> | -                 | ICL+RA      | **43.32%**   | 6.42%     | **58.82%** | **10.96%** | **9.37%**  | 1.87%     | 15.19%     | 3.12%      |
> | -                 | ICL+RA (+R) | 61.50%       | 17.91%    | 72.99%     | 20.32%     | 48.18%     | 32.36%    | 62.02%     | 41.94%     |
>
> For GPT-4o, the results meet our expectations: RA consistently improves all metrics on all benchmarks (except BEq@1 on ProofNet), and RA(+R) shows the potential of dependency retrieval.
> However, for Deepseek-V2.5, RA doesn't work well. We hypothesize this might result from insufficient instruction-following and long-context capabilities of Deepseek-V2.5. The noise in retrieved dependencies degrades autoformalization. But RA (+R) shows significantly better performance as expected.
>
> Thank you again for this interesting idea. This discussion is also added to _Appendix A.10_.
>
> [1] https://ammkrn.github.io/type_checking_in_lean4/type_checking/definitional_equality.html
>
> [2] https://ammkrn.github.io/type_checking_in_lean4/type_checking/reduction.html#nat-literal-reduction

---

> ### Comment · Area_Chair_hUz9 · 2024-11-25
> **Reminder: Rebuttal Deadline for ICLR 2025**
>
> Dear Reviewer fV7x,
>
> As the rebuttal deadline approaches, please kindly check the papers' discussion threads and respond to the authors' rebuttals. If you haven't had a chance to respond yet, I’d greatly appreciate your input soon. Your insights are invaluable to the authors and the review process.
>
> Thank you for your effort and support!
>
> Best regards,
>
> Area chair

---

> ### Comment · Reviewer_fV7x · 2024-12-03
> **Thanks for your responses**
>
> The response satisfactorily addresses my concerns.  I thank the authors for their detailed response and effort during the rebuttal!

---

### Official Review · Reviewer_pPF9 · 2024-11-04

**Soundness:** 3
**Presentation:** 3
**Contribution:** 4
**Rating:** 6
**Confidence:** 3

**Summary:**

The work attempts to address two important gaps in statement autoformalization currently: 1) absence of an automated and accurate evaluation metric for autoformalization; 2) models' agnosia of contextural information during autoformalization, inducing severe hallucination of formal definitions and theorems.

- To address 1), the paper proposes BEq (Bidirectional Extended Definitional Equivalence), a neural-symbolic metric that determines the equivalence between an autoformalized statement and the ground-truth formal statement by taking advantage of Lean 4 features.

- For 2), the paper introduces RAutoformalizer (Retrieval-augmented Autoformalizer) which (a) uses dependency retrieval to find relevant formal objects from libraries, then (b) proposes topological informalization to structure training data. RAutoformalizer significantly improves performance on ProofNet and the out-of-distribution benchmark Con-NF.

- The authors also synthesized Con-NF, a new benchmark with 961 informal-formal statement pairs, by informalizing from a frontier mathematical library that has not been adopted before, to evaluate OOD generalization capabilities in autoformalization.

**Strengths:**

1. **Originality**:

- BEq is one of the pioneer metrics for evaluating autoformalization different from existing ones. It is built based on human intuitions about statement equivalence and implemented using Lean 4 features, thus having the potential advantage of being more interpretable and predictable (although not elaborated in the paper), if provided with high-quality ground-truth formal statements.
- The paper also formulates the problem of autoformalization as bidirectional definitional equivalence. This approach differs from previous formulation efforts (e.g., https://arxiv.org/pdf/2410.04194).
- RAutoformalizer introduces a typological informalization and dependency retrieval pipeline to reduce context agnosia. The Con-NF benchmark also emphasizes the importance of OOD evaluation for future researchers.

2. **Quality**:
  - Uses both in-distribution (ProofNet) and out-of-distribution (Con-NF) testing;
  - Includes ablation studies to validate technical choices;
  - Provides training details for RAutoformalizer.

3. **Clarity**: The motivations are coherent. The writing and illustrations are mostly clear. However, improvements can be made regarding the formal problem setup when introducing BEq as well as the implementation procedures.

4. **Significance**: The paper aims to address two important problems in autoformalization that are both relevant and challenging. The significance is very valid, although the applicability of BEq is compromised as mentioned later.

**Weaknesses:**

BEq:

1. **Limited applicability**:
   - A limitation of BEq for evaluating autoformalization is it focuses on the matching between ground-truth formal statement and the autoformalized statement, hence it always requires ground-truth formal statements. In other words, BEq disregards the informal statement input and relies heavily on the *quality* of the ground-truth formal statement that the human experts provided. It also relies on the tactic features of Lean 4. This greatly hurts the applicability of the metric to many larger-scale informal-formal language benchmarks without expert-annotated ground-truths or written in other formal languages. It also makes it questionable whether BEq can even be directly used on synthesized benchmarks like Con-NF where the informal-formal statement pairs are not checked by human experts.
   - The weakness should be also be mentioned in the theoretical constraint of the problem formulation in section 3. Fundamentally, BEq distills the problem of evaluating autoformalizatiom into the problem of evaluating 'Definitional Equality' between two formal statements in Lean 4. However, this overlooks two constraining assumptions that should be ensured about the autoformalization benchmarks: (1) the ground-truth formal statements are aligned with the informal statements; (2) the ground-truth formal statements are successfully type-checked.

2. **Limited robustness**: There are few experiment details about how human evaluation are conducted such as the expert numbers, competence levels, annotation protocols, annotation processes, agreement rates, etc. Considering that evaluating formal statements is a challenging task that frequently induces high disagreement rates among experts in practice, the rigor of BEq and Table 1 results is in question.
3. **Unclear experimental setting missing material support**: The paper also did not provide concrete experimental details about Beq such as (a) the few-shot prompt for LLMs on transformation T (line 261), (b) the few-shot prompt for LLMs graders on formal statement matching (line 303). The authors cited Ying et al. (2024a) and stated they used a stronger few-shot setting of their prompt, but I did not find the corresponding prompt template in the reference. It is difficult to review the exact implementation process and experimental details, hurting the robustness and reproducibility of the work.
4. **Writing clarity issue**: The formal setup of BEq is unclear. Please check point 1 and 2 in Questions for details.

RAutoformalizer:

5. **Benchmark construction rigor**: The OOD benchmark Con-NF is constructed by prompting LLMs to typologically informalize the formal objects in a new library. There is little quality control as to the informalization quality. This is especially concerning since we are synthesizing data from a new library that LLMs are probably never exposed to. Two issues may arise from this that have been noted in previous research:
   - There could be a great discrepancy between the informal statements in Con-NF that serve as autoformalization input and the groundtruth formal statements from the first place.
   - In LLM informalization via prompting, a common issue is that the informalized output often still includes the names of the formal tactics or variables verbatim, because there does not exist a corresponding concept in the natural language that can be used to easily translate the complex functions or relationships in formal language. In more extreme cases, some theorems are not even translatable formally or informally. The paper did not report any filtering procedures or quality checks regarding the potential benchmark issues above.

6. **No human evaluation in comparing RAutoformalizer**: Table 3 compares the autoformalization performance of RA with other baselines on ProofNet and OOD Con-NF using two automated metrics: Typecheck and BEq. Because the robustness of BEq itself is limited as discussed above, the significance of the table results is compromised unless human evaluations are provided.

**Questions:**

1. I am not clear what the notion ‘transformation primitives’ in line 215/ line 221 means: in some in autoformalization research, 'transformation' is sometimes an equivalent of 'autoformzalization', but here apparently it is not the case. Does 'all transformation primitives' here refer to all functions defined in Definitional Equality in Lean 4 (e.g., α-conversion, η-expansion etc.)? The confusion about transformation T continued regarding how LLMs are prompted to implement the transformation function T. The paper would be clearer and more reproducible if the authors provide more accurate declarations of each symbol and notion, as well as the exact prompt template they use for few-shot prompting. Another example of ambiguous symbol usage is ‘s\_P’ versus ‘theorem P’.

2. Following the question about T, I wonder how the authors derive the Unidirectional Definitional Implication (2) from equation (1) by explaining how transformations can occur between a formal statement and a proof goal in Lean 4 around line 225-236. Is the T in (2) an approximation of T in (1), based on the fact we can switch between proof goals and formal statements in Lean 4?

3. A recent work called FormalAlign introduces a metric for autoformalization evaluation to address the first limitation in this paper by training a *formal-informal statement alignment* evaluator, rather than focusing on *formal-formal equivalence*: [https://arxiv.org/pdf/2410.10135v1](https://arxiv.org/pdf/2410.10135v1). The authors are suggested to discuss this in the related works section and ideally include a comparison with this metric as a baseline.

4. Typo and grammatical issue:
   - Line 097 (‘A baseline is build’); line 199 (‘with how humans instincts.’);
   - In Table 3, in the Con-NF / Typecheck@1↑ column, LW achieves the highest (28.10%) but RA is marked bold (20.50%). If there is indeed a mistake here, the body text should also be adjusted.

---

> ### Author Response · Authors · 2024-11-22
>
> Many thanks for your constructive comments. Responses to the aforementioned weaknesses and questions are as follows. We sincerely hope them to resolve the concerns.
>
> > Weakness1. Potential limitations of BEq.
> > 1. BEq focuses on comparing model prediction to ground-truth formal statement, which requires ground-truths.
> > 2. BEq relies on the tactic features of Lean 4.
> > 3. Evaluation of BEq on synthesized benchmarks like Con-NF is unreliable.
> > 4. BEq distills the problem of evaluating autoformalization into the problem of evaluating 'Definitional Equality' between two formal statements in Lean 4.
>
> We beg to differ with some of your comments.
>
> 1. Admittedly, this is one of the biggest limitation of BEq, as well as an **open problem of the whole ML community** for years. It is difficult to evaluate a task of the form $\boldsymbol y=f(\boldsymbol x)$ without a ground-truth $\boldsymbol y$. All discriminative tasks need to compare predictions with gold ground-truth labels. For generative tasks, to coarsely summarize, there are mainly 4 types of automated evaluation methods:
>     1. Using learning-based metrics, such as FID (Fréchet inception distance) in image generation tasks;
>     2. Using symbolic metrics, such as calculator or formal verifier (in theorem proving);
>     3. Partial evaluation. For example, existing informal reasoning benchmarks such as MATH[3] and MMLU[2] only perform identity matching between model-predicted answers and the ground-truth answers, neglecting the reasoning trajectory;
>     4. Proxy metrics, such as BLEU in machine translation.
>
>     1,3,4 either depend on ground-truth labels or an “evaluator” DNN, which is uninterpretable and vulnerable to adversarial attacks. And 2 is not suitable for our task, because the space of free-form natural language statements in autoformalization is intractable for symbolic verifier.
>     We currently have no idea of this limitation and believe solving this can lead to strong contribution for the whole ML community.
>     As for autoformalization, **all existing benchmarks (e.g. ProofNet[1], FormL4[9]) have ground-truth formal statements**. And it should be the benchmark authors' responsibility to ensure the quality of ground-truth.
> 2. The idea of BEq, which defines an equivalence relation as *bidirectionally convertible under restricted transformations*, can be applied in broader areas. Our implementation of BEq on Lean relies on two key components: definitional equality and tactics. Definitional equality is a widespread concept in formal verification environments, especially dependent-type-theory-based languages such as Coq and Lean; Tactics are intrinsically metaprograms that transform the current goal, which can also be replaced by similar concepts in other formal verification environments.
> 3. Thank you for pointing out this issue. We have conducted comprehensive human evaluation on the autoformalization results of multiple settings (including Con-NF) and added it as **Appendix A.2.1**. Results show that BEq even better aligns with human experts on Con-NF.
>
> | Benchmark | Method | TP | TN | FP | FN | Precision | Recall  | Accuracy |
> |-----------|--------|----|----|----|----|-----------|---------|----------|
> |  ProofNet | RA -R  | 22 | 70 | 0  | 9  | 100.00%   | 70.97%  | 91.09%   |
> |           | RA     | 22 | 67 | 0  | 11 | 100.00%   | 66.67%  | 89.00%   |
> |           | RA +R  | 32 | 57 | 0  | 12 | 100.00%   | 72.73%  | 88.12%   |
> |   Con-NF  | RA -R  | 29 | 49 | 0  | 0  | 100.00%   | 100.00% | 100.00%  |
> |           | RA     | 55 | 44 | 0  | 1  | 100.00%   | 98.21%  | 99.00%   |
> |           | RA +R  | 74 | 23 | 0  | 3  | 100.00%   | 96.10%  | 97.00%   |
>
> 4. We respectfully disagree with you as evaluating autoformalization results between model prediction and ground-truth is a mainstream paradigm adopted by perplexity, BLEU, E3[4]. Only two exceptions exist: 1) Typecheck, which only checks whether Lean kernel can successfully compile and totally neglects semantic difference. 2) LLM Grader[5], which is NOT used as an evaluation method but a filtering method in the original paper.
>
> 	We believe it should be the benchmark authors' responsibility to ensure the quality of ground-truth, including semantic correctness (the ground-truth formal statements are aligned with the informal statements) and syntactic correctness (the ground-truth formal statements are successfully type-checked).

---

> ### Author Response · Authors · 2024-11-22
>
> > Weakness2. Potential quality issue of Human Equivalence Benchmark.
>
> Thank you for your careful review. More details about the construction of human equivalence benchmark have been added in our revised version in **Appendix A.4 Human Equivalence Benchmark**. The expert group consists of 1) one professor in formal verification; 2) one PhD student in EECS; 3) two master students in CS. However, In accordance with the principles of anonymity, we regret to inform you that we cannot disclose any additional information. As stated in Sec.7 Reproducibility Statement, the Human Equivalence Benchmark will be released after acceptance and whoever interested can check then.
>
> > Weakness3. Unclear prompt details of BEq and LLM Grader.
>
> Thank you again for your careful review. We have added these details in our revised version in Appendix A.7.
>
> > Weakness4. Writing clarity.
>
> Many thanks for the advice. We have revised all writing clarity issues in the main text and Fig.1.
>
> > Weakness5. Construction of Con-NF benchmark
> > 1. Lack of quality control
> > 2. Discrepancy between the informal statements and ground-truth formal statements
> > 3. Possibility of the informal statements to contain formal identifiers or even raw formal statements
>
> 1. Rejective sampling and post-informalization filtering policies are applied in benchmark construction, including
> 	1. Length filtering: Generated informal statements which are too long or too short are rejected.
> 	2. Format filtering: Generated informal statements which do not match the given format are rejected.
> 	3. Special token filtering: Generated informal statements which contain some special tokens (which we think to be a signal of low-quality informalization) are rejected.
> 	4. Semantic filtering: Sophisticated rules are applied to filter out samples with large semantic gaps with the corresponding math object.
> 	5. Source filtering: Samples whose all dependencies are in Mathlib 4 are filtered out to ensure the OOD nature of this benchmark.
>
> 	Moreover, human supervision is conducted on statements that fail to pass the filters after several sampling attempts. Post-informalization filtering reduces the size of the Con-NF benchmark from 85762 to 961.
>
> 2. Thank you for the comment. Human evaluations are conducted on 20 samples of informal-formal pairs. The results indicate that 2 of them show obvious semantic discrepancies, 9 of them contain raw formal identifiers, and none of them failed to translate. We must admit that there is some noise in the benchmark. However, this noise is significantly smaller than the $\ge 25\\%$ noise found in ProofNet[8]. The conclusion of RAutoformalizer's ablative improvement is not influenced because we can observe that RAutoformalizer (-R) and all baselines show similar low accuracy on Con-NF ($\le 4.58\\%$), while RAutoformalizer and RAutoformalizer (+R) have substantial improvements ($\ge 16.86\\%$).
>
> 3. This possibility does exist. As in (2.), we find 9 out of 20 samples contain formal identifiers in the informal statements. However, we don't think these formal identifiers might make the evaluation inaccurate or unfair. As the results show, all non-retrieval methods, including RAutoformalizer (-R), have significantly lower accuracy than retrieval-augmented ones ($\ge 10\\%$ absolute improvement).
>
> > Weakness6. Human evaluation in comparing RAutoformalizer.
>
> Thank you for your suggestion. We have conducted human evaluation for RAutoformalizer and its ablative variants (-R, +R) on ProofNet and Con-NF, respectively. For each experiment, we evaluate about 100 samples from generated formal statements that passed typecheck. Since the number of type-checked generations from RAutoformalizer (-R) was fewer than 100, we evaluated all of them. Please refer to **Appendix A.2.2**, where the human-rectified accuracies more strongly supports the ablative improvement of Dependency Retrieval. The comparison between human-annotated equivalence labels and BEq results (as shown in **Appendix A.2.1**) also show BEq's high-fidelity in alignment with human experts.

---

> ### Author Response · Authors · 2024-11-22
>
> > Question1. Writing clarity:
> > - What the notion ‘transformation primitives’ in line 215/ line 221 means?
> > - How the authors derive the Unidirectional Denitional Implication (2) from equation (1)?
> > - Ambiguous symbol usage is 's_P' versus 'theorem P'
>
> Sorry for causing your misunderstanding. We have polished our writing in the revision. Here are some explanation of terminology and symbol.
> - "Transformation primitives" indeed refers to tactics in Lean (cf. Line 233 "concretize $\mathbb R$ to be the set of all tactics in Lean").
> - $T(\boldsymbol s_P | \boldsymbol s_Q, \mathcal R)$ is an oracle function that given a target statement $\boldsymbol s_Q$ and a set of available transformation primitives $\mathcal R$, it applies primitives in $\mathcal R$ and tries to transform $\boldsymbol s_P$ to be definitionally equal to $\boldsymbol s_Q$. If unable to transform (the semantics discrepancy between $\boldsymbol s_p$ and $\boldsymbol s_Q$ are too large or restriction on $\mathcal R$ is too tight), it returns a dummy placeholder. In our implementation, to approximate this oracle function, we prompt a LLM to try to generate a transformation from $\boldsymbol s_P$ to $\boldsymbol s_Q$. More concretely: firstly, assume $\boldsymbol s_Q$ to be true (by closing it with `sorry`). Then we sample proofs from a few-shot prompted LLM for proofs of $\boldsymbol s_P$ which should use $\boldsymbol s_Q$ and be composed of the tactics in $\mathcal R$.
> - "theorem P" refers to the semantics of a theorem, while $\boldsymbol s_P$ refers to its formal statement (or more precisely, one of the elements in the equivalence class of theorem P in human perspective).
>
> > Question2. Is the T in (2) an approximation of T in (1)?
>
> No, they mean the same oracle function. Please refer to Answer Point 2 to Question 1 for more explanation.
>
> > Question3. Comparison to FormalAlign
>
> An intriguing question. It is worth noting that [FormalAlign](https://openreview.net/forum?id=B5RrIFMqbe) is also an ICLR submission, and its Arxiv date is later than ICLR submission date, so we will not include it into the main text. However we still provide a comparison to satisfy readers' curiosity (and also ours).
> Actually, FormalAlign addresses a different task compared to BEq. FormalAlign aims at measuring the alignment between **a formal statement and an informal statement**, while BEq is used to determine the equivalence between **two formal statements**.
> Since FormalAlign's models, training code and inference code are all unavailable, it is quite difficult to evaluate on Human Equivalence Benchmark. Fortunately, the authors have published their synthesized benchmarks, on which we can evaluate BEq.
> For a fair and comprehensive comparison, many efforts are made, including extracting formal statements from contaminated text (seems due to incorrect response from their GPT calling), identifying imports and namespaces, filtering out statements that do not pass the typecheck, fixing broken tokens stemming from over-augmentation, etc.
>
> Finally, we evaluate BEq($k=4$)'s capabilities to discriminate misaligned statements with their ground-truths (`misalignment`). The results are as follows.

---

> ### Author Response · Authors · 2024-11-22
>
> | **Datasets**      | **Error Type** | **Total** | **Error** | **Precision** | **Precision (FormalAlign)** |
> | ----------------- | -------------- | --------- | --------- | ------------- | --------------------------- |
> | **FormL4-Basic**  | equality       | 148       | 4         |               |                             |
> |                   | unpaired       | 200       | 0         |               |                             |
> |                   | constant       | 237       | 0         |               |                             |
> |                   | variable_new   | 1         | 0         |               |                             |
> |                   | _Total_        | _586_     | _4_       | **99.32%**    | 93.65%                      |
> | **FormL4-Random** | constant       | 334       | 7         |               |                             |
> |                   | equality       | 103       | 0         |               |                             |
> |                   | unpaired       | 276       | 0         |               |                             |
> |                   | variable_new   | 353       | 313       |               |                             |
> |                   | variable_type  | 23        | 0         |               |                             |
> |                   | exponent       | 28        | 0         |               |                             |
> |                   | _Total_        | _1117_    | _320_     | 71.35%        | **86.90%**                  |
> | **MiniF2F-Valid** | constant       | 1469      | 32        |               |                             |
> |                   | variable_type  | 117       | 1         |               |                             |
> |                   | equality       | 231       | 0         |               |                             |
> |                   | exponent       | 288       | 9         |               |                             |
> |                   | unpaired       | 1150      | 1         |               |                             |
> |                   | _Total_        | _3255_    | _43_      | **98.68%**    | 68.58%                      |
> | **MiniF2F-Test**  | constant       | 1491      | 49        |               |                             |
> |                   | variable_type  | 116       | 0         |               |                             |
> |                   | equality       | 226       | 0         |               |                             |
> |                   | exponent       | 255       | 5         |               |                             |
> |                   | _Total_        | _2088_    | _54_      | **97.41%**    | 66.70%                      |
>
> We find the following error patterns of BEq on this benchmark.
> 1. (`constant`, `exponent`, `unpaired`) Samples marked error in this category modify natural number constants, but the modified expression has the same value as the original one. This might stem from two reasons.
> 	1. The modified number reduces to the same value in type conversion. For example, changing `{n : ℕ} : (n : ZMod 2) = 0 ↔ Even n` to `{n : ℕ} : (n : ZMod 2) = 44 ↔ Even n`. When Lean instantiates this statement, it firstly converts the natural number $44$ into type `ZMod 2`, where it reduces to `0 : ZMod 2`.
> 	2. The modified expression is composed of natural numbers and natural number operators, and reduces to the same value. For example, changing `(29 * 79 + 31 * 81) % 10 = 2` to `(29 * 89 + 31 * 81) % 10 = 2`. Because both left hand sides `(29 * 79 + 31 * 81) % 10` and `(29 * 89 + 31 * 81) % 10` reduces to `2` in Nat literal reduction[6], they are definitionaly equal in Lean.
> 2. (`variable_new`) Samples marked error in this category introduce new variables of nonempty types, but they are not incorporated in any hypothesis or the goal statement. For example, changing `{p : α → Prop} {x : α} : { x | p x } x ↔ p x` to `{p : α → Prop} {x k : α} : { x | p x } x ↔ p x`. The two statements are equivalent under BEq since they can prove each other by adding a dummy parameter or omitting a parameter. Significantly, if the introduced variables are of some empty type, the modified statements are not equivalent anymore, e.g. `(x : Nat) : x + 1 > x` are inequivalent to `(x : Nat) (hN : False) : x + 1 > x`.
> 3. (`equality`): Samples marked error in this category change statements like `a * b = b ↔ a = 1` to `a * b ≠ b ↔ a ≠ 1`. These two expressions are trivially logically equivalent.
> 4. (`variable_type`): Samples marked error in this category change expressions like `((2 : ℝ)^3 + Real.sqrt 9)` to `((2 : ℚ)^3 + Real.sqrt 9)`. In the latter, `(2 : ℚ)` is finally converted to `2 : ℝ` due to the subsequent addition with `(Real.sqrt 9) : ℝ`. Therefore, these two expressions are definitionally equal in Lean 4.
>
> Note that this comparison is slightly unfair because these two methods attends to different tasks. And hopefully, the above analysis can inspire the readers.

---

> ### Author Response · Authors · 2024-11-22
>
> > Question4. Typos.
>
> Many thanks for your careful review. These typos and grammatical issues are fixed in our revision.
>
> [1] Azerbayev, Zhangir, et al. "Proofnet: Autoformalizing and formally proving undergraduate-level mathematics." _arXiv preprint arXiv:2302.12433_ (2023).
>
> [2] Hendrycks, Dan, et al. "Measuring massive multitask language understanding." _arXiv preprint arXiv:2009.03300_ (2020).
>
> [3] Hendrycks, Dan, et al. "Measuring mathematical problem solving with the math dataset." _arXiv preprint arXiv:2103.03874_ (2021).
>
> [4] Murphy, Logan, et al. "Autoformalizing Euclidean Geometry." _arXiv preprint arXiv:2405.17216_ (2024).
>
> [5] Ying, Huaiyuan, et al. "Lean Workbook: A large-scale Lean problem set formalized from natural language math problems." _arXiv preprint arXiv:2406.03847_ (2024).
>
> [6] https://ammkrn.github.io/type_checking_in_lean4/type_checking/reduction.html#nat-literal-reduction
>
> [7] https://ammkrn.github.io/type_checking_in_lean4/type_checking/definitional_equality.html#eta-expansion
>
> [8] https://leanprover.zulipchat.com/#narrow/channel/219941-Machine-Learning-for-Theorem-Proving/topic/ProofNet.20fix
>
> [9] Lu, Jianqiao, et al. "Process-driven autoformalization in lean 4." arXiv preprint arXiv:2406.01940 (2024).

---

> ### Comment · Area_Chair_hUz9 · 2024-11-25
> **Reminder: Rebuttal Deadline for ICLR 2025**
>
> Dear Reviewer pPF9,
>
> As the rebuttal deadline approaches, please kindly check the papers' discussion threads and respond to the authors' rebuttals. If you haven't had a chance to respond yet, I’d greatly appreciate your input soon. Your insights are invaluable to the authors and the review process.
>
> Thank you for your effort and support!
>
> Best regards,
>
> Area chair

---

> ### Comment · Reviewer_pPF9 · 2024-11-25
> **Reviewer Response to Authors' Comments**
>
> I thank the authors for their detailed response. Some of the concerns or disagrements remain as follows.
>
> ---
>
> **Weakness 1**:
>
> **W1.1.**
> > It is difficult to evaluate a task of the form y=f(x) without a ground-truth y. All discriminative tasks need to compare predictions with gold ground-truth labels. For generative tasks, to coarsely summarize, there are mainly 4 types of automated evaluation methods: ...
>
> This claim is inaccurate. Generation evaluation metrics do not always require ground-truths. In theory, an ideal evaluation can independently assess a task-specific generation without given ground-truths, as long as the task objective is clearly defined. In practice, model-based "evaluator" or LLM-as-judge can both be reference-free. Listing their limitations such as "uninterpretable" does not mean they can be simply disregarded. Rather, some neural networks or LLMs perform extremely well and robustly in evaluation tasks.
>
> Regarding the paper itself, I have two concerns:
>
> 1. I agree that "it should be the benchmark authors' responsibility to ensure the quality of ground-truth". So you can say that it is more like a limitation in methodology rather than a fatal flaw. However, it should be highlighted to the readers Beq's assumption on dataset quality when introducing the metric in the paper (e.g., in section 3). Large-size datasets like Forml4, MMA are not 100\% manually verified that the informal and formal statements are perfectly aligned, so they are actually not applicable.
>
> 2. The authors mentioned the limitation in L518-522. However, based on the points above, I think the subsequent line is far too absolute and defensive, stating "This limitation is unavoidable throughout the machine learning community".
>
> **W1.2.**
>
> The logic link from evaluating formal statement equivalence (your metric) to evaluating autoformalization (your motivation) is still missing. E.g.,. L101 says "identify two key limitations in **statement autoformalization: the absence of faithful and universal automated evaluation**...", while Beq is defined as "a neuralsymbolic **equivalence relation between formal statements**" (L77). I think more clarification should be made early in the paper to stress your assumption to transform the problem of evaluating informal-formal statement alignment into evaluating equivalence between generated and ground-truth formal statements. Admittedly, it is a mainstream paradigm, so you can make analogies to e.g., the evaluation metrics in cross-lingual machine translation.
>
>
> ---
>
> **Other Weaknesses**:
>
> Thank you for the detailed responses, especially the human evaluation results and the inspiring comparison with FormalAlign.

---

> > ### Author Response · Authors · 2024-11-26
> >
> > Thank you again for your rigorous and constructive comment. We sincerely hope the following explanations can help resolve your concerns.
> >
> > > W1.1. Inaccurate statement that "It is difficult to evaluate a task of the form y=f(x) without a ground-truth y."
> >
> > We wholeheartedly admit that this sentence was misleading.
> >
> > Here we were simply elucidating the thoughts underlying our preliminary task design: in our humble opinions, what the community urgently needs is a faithful and interpretable metric to automate the evaluation of statement autoformalization. Among multiple design choices, we follow prevalent benchmarks such as ProofNet[1], and LeanEuclid[2] to evaluate autoformalization by comparing with ground-truths.
> >
> > Our intention was not to assert that generation evaluation metrics invariably necessitate ground-truths. As detailed in the "4 types of automated evaluation methods for generative tasks", methods within types 1 (learning-based metrics), 3 (partial evaluation), and some of 4 (proxy metrics) are precisely the generation evaluation metrics that can function without ground-truths.
> >
> > Furthermore, we **in no way meant to downplay the importance of model-based evaluators**. Conversely, we think these methods, such as FID[3] in image generation, reward modeling, and FormalAlign[4] are excellent evaluation metrics as they successfully use data-driven training to mimic human preference, and can be potentially used in rejective sampling to enhance generation quality.
> >
> > We apologize for any confusion this statement may have caused and hope this explanation clarifies our position.
> >
> > > W1.1, Concern 1. Highlighting the limitation of BEq in the paper
> >
> > Glad to see that we have reached a consensus that "it should be the benchmark authors' responsibility to ensure the quality of ground-truth" :)
> > As for the limitation of requiring ground-truth and the assumption on benchmark quality, please kindly refer to **Sec.6.Limitations of BEq**, where it's already highlighted at the very beginning.
> >
> > > W1.2, Concern 2. Discussion of limitation is too defensive.
> >
> > Thank you for pointing out this confusion. In the new revision, the limitation of BEq is rewritten as follows:
> >
> > *As an equivalence metric between formal statements, the accuracy of BEq depends on the quality of the ground-truth formal statements of the benchmarks. Therefore, BEq is not suitable for benchmarks with low-quality ground-truths or those lacking formal ground-truths.*
> >
> > Hopefully, this emphasis can be more objective and helpful for the readers.
> >
> > > W1.2. Absence of logical link from evaluating autoformalization to comparing with ground-truth formal statement.
> >
> > We appreciate your insightful suggestion for the writing coherence. The introduction and motivation of BEq are rewritten (Sec.3, L181-194), adding the explanation of evaluating autoformalization by comparing predictions and ground-truths, and a more detailed motivation of BEq. The explanation is expressed as *"In statement autoformalization, we follow prevalent benchmarks such as ProofNet (Azerbayev et al., 2023) and LeanEuclid (Murphy et al., 2024) to evaluate by comparing model predictions with ground-truths"*.
> >
> > Due to the page limit of the paper, we're afraid we can't afford to elaborate on the implicit assumptions and limitations here in more detail. These aspects have already been thoroughly highlighted in **Sec.6.Limitation and Broader Impacts**.
> >
> > Hopefully, this revision will address your concerns.
> >
> >
> > [1] Azerbayev, Zhangir, et al. "Proofnet: Autoformalizing and formally proving undergraduate-level mathematics." _arXiv preprint arXiv:2302.12433_ (2023).
> >
> > [2] Murphy, Logan, et al. "Autoformalizing Euclidean Geometry." _arXiv preprint arXiv:2405.17216_ (2024).
> >
> > [3] Heusel, Martin, et al. "Gans trained by a two time-scale update rule converge to a local nash equilibrium." _Advances in neural information processing systems_ 30 (2017).
> >
> > [4] Lu, Jianqiao, et al. "FormalAlign: Automated Alignment Evaluation for Autoformalization." _arXiv preprint arXiv:2410.10135_ (2024).

---

> > > ### Comment · Reviewer_pPF9 · 2024-11-28
> > >
> > > I appreciate the author's responses. The revisions you mentioned have strengthened the paper's clarity and improved its narrative. I have raised the sub-scores of `soundness` to 3.

---

> > > > ### Author Response · Authors · 2024-11-29
> > > >
> > > > Dear Reviewer pPF9,
> > > >
> > > > We are extremely grateful for your careful review and valuable feedback. It is delightful to know that the revisions have enhanced the paper's clarity and narrative, and we also appreciate that you have adjusted the soundness score upward.
> > > >
> > > > We were just wondering if there might be any other remaining concerns. If, by any chance, there aren't, would it be possible for you to take into account reconsidering the overall rating? Your evaluation is of great significance to us, and we truly appreciate your efforts.
> > > >
> > > > Thank you again for your time and thoughtful consideration. We are earnestly awaiting your response.
> > > >
> > > > Best regards,
> > > >
> > > > Authors

---

### Public Comment · ~Auguste_Poiroux1 · 2024-11-29
**Intriguing metric results**

When looking at your reported metrics on ProofNet, I noticed a strange issue. ProofNet benchmark has two splits (valid and test) of sizes 185 and 186 [1]. The BEq metric you present is a binary metric, so when evaluating, you should get an integer value corresponding to the number of “correct” predictions (in your paper, equation (7) multiplied by N). However, for some of your percentage results when I try to multiply them by any of the splits size (185 and 186), or by the total size of the benchmark (371), I don't get integer values. For example, if we try the value reported in your abstract, 12.83%, we get:

12.83% * 185 = 23.7355 (closer possible values are 23/185=12.43% and 24/185=12.97%) → conclusion: not evaluated on the valid set

12.83% * 186 = 23.8638 (closer possible values are 12.37% and 12.90%) → conclusion: not evaluated on the test set

12.83% * 371 = 47.5993 (closer possible values are 12.67% and 12.94%) → conclusion: not evaluated on the valid + test set

And this happens for various reported values. According to this remark, it seems that you used a custom subset of ProofNet and not the full set. However, for PDA method, you report 0.27% which corresponds to exactly 1 correct prediction over 371 (1/371 = 0.27%). Since ProofNet has a total size of 371, this would mean that you used the full ProofNet dataset. These contradicting facts are a bit confusing, what is happening here? Am I missing something? Could you clarify how you performed the evaluation on ProofNet?

[1] https://huggingface.co/datasets/hoskinson-center/proofnet

---

> ### Author Response · Authors · 2024-11-29
>
> Good question! In fact, the published version of ProofNet [1] contains 374 informal-formal statement pairs (with 3 from `Cambridge-Tripos.lean`, which was not mentioned in the original paper). The Lean 4 version of ProofNet we adopted ([2], Table 9 in our submission) also contains 374 problems.
>
> We evaluate on the whole benchmark, where $12.83\\% \approx \frac{48}{374}$
>
> Actually, there was a footnote in the initial submission explaining this issue: "_The published version of ProofNet (Azerbayev et al., 2023) consists of 374 informal-formal statement pairs of undergraduate-level mathematics_".
> However, in the subsequent revisions, since we needed to add more important content and to make room for that, we removed this footnote.
> We apologize for the misunderstandings and confusions caused by this.
>
> [1] https://github.com/zhangir-azerbayev/ProofNet/tree/main/benchmark/benchmark_to_publish/formal
>
> [2] https://github.com/rahul3613/ProofNet-lean4/tree/main/formal

---

> > ### Public Comment · ~Auguste_Poiroux1 · 2024-12-01
> >
> > Interesting, thank you for this clarification!

---

### Meta-Review · Area_Chair_hUz9 · 2024-12-19

**Metareview:**

Summary of the paper:
This paper focuses on two main limitations of autoformalization: the lack of an effective automated evaluation metric and the models' inability to utilize contextual information, leading to inaccuracies in formal definitions and theorems. To tackle these issues, the authors propose BEq, a neural-symbolic metric designed to evaluate the equivalence between formalized statements and their ground-truth counterparts, leveraging features from Lean 4. Additionally, the authors introduce RAutoformalizer, which employs dependency retrieval to source relevant formal objects and utilizes topological informalization to enhance training data organization. This method shows significant performance improvements on ProofNet and a new out-of-distribution benchmark called Con-NF, which comprises 961 informal-formal statement pairs derived from a novel mathematical library. Experiments show that BEq provides expert-level performance in assessing semantic equivalence, while RAutoformalizer outperforms existing state-of-the-art approaches.

Strengths of the paper:
- Originality of BEq Metric: Introduces a pioneering metric for evaluating autoformalization based on the bidirectional definitional equivalence, grounded in human intuitions about statement equivalence and utilizing Lean 4 features for interpretability, improving alignment with human evaluations.
- RAutoformalizer Pipeline: Implements a typological informalization and dependency retrieval pipeline that mitigates context agnosia, enhancing the autoformalization process by ensuring contextual fluency for advanced mathematical concepts, and demonstrating superior experimental results.
- New Benchmark Creation: Establishes the Con-NF benchmark from mathematical research, emphasizing the importance of out-of-distribution evaluations.

Weaknesses of the paper:
- Clarity and Organization (Reviewer pPF9, 6DKY): In general, the paper is well-written and logically structured, though some areas could benefit from more clarity in the formal problem setup.
- Potential limitations of BEq (Reviewer pPF9, 9PuK): It requires available ground-truth formal statements and relies on specific Lean 4 tactic features, which may restrict its applicability. Its evaluation on synthesized benchmarks, like Con-NF, is deemed unreliable, and it simplifies the autoformalization assessment by reducing it to 'Definitional Equality' between formal statements, potentially overlooking the evaluation's complexity. Lastly, BEq requires a 20B parameter model to execute.
- Potential quality issue of Human Equivalence Benchmark (Reviewer pPF9): The benchmark Con-NF is constructed by prompting LLMs to typologically informalize the formal objects in a new library. There is little quality control as to the informalization quality. This is especially concerning since we are synthesizing data from a new library that LLMs are probably never exposed to.
- More experimental results are needed (Reviewer fV7x, 9PuK): Including more ablative analysis of RAutoformalizer, stronger Dependency-retrieval-augmented ICL baselines, existing reasoning baselines and more implementations of different families of LLMs with BEqs.

Reasons for the decision: After considering the rebuttal, I believe the authors have adequately addressed most of the concerns raised, as discussed in the reviewer discussion below. All reviewers agree that this paper should be accepted at ICLR. Upon careful reflection, I find that this paper makes a significant contribution to the autoformalization and formal mathematics research community. It introduces a more reliable evaluation metric, an innovative autoformalization technique, and a benchmark for measuring out-of-distribution generalization in this domain. These contributions tackle critical challenges in the field effectively.

**Additional Comments On Reviewer Discussion:**

The authors address most of the concerns raised by the reviewers:
- Clarity and Organization (Reviewer pPF9, 6DKY): The authors provide further necessary information in the Appendix.
- Potential limitations of BEq (Reviewer pPF9, 9PuK): The authors provide more contexts and additional experiments to clarify the limitations.
- Potential quality issue of Human Equivalence Benchmark (Reviewer pPF9): The required information is added in the Appendix.
- More experimental results are needed (Reviewer fV7x, 9PuK): All experiments suggested are conducted and attached during the rebuttal.

---

### Decision · Program_Chairs · 2025-01-22

Accept (Spotlight)